

# Sources of organic aerosols in Europe: A modelling study using CAMx with modified volatility basis set scheme

Jianhui Jiang[1], Sebnem Aksoyoglu[1], Imad El-Haddad[1], Giancarlo Ciarelli[2], Hugo A. C. Denier van der Gon[3], Francesco Canonaco[1], Stefania Gilardoni[4], Marco Paglione[4,*], María Cruz Minguillón[5], Olivier Favez[6], Yunjiang Zhang[6,7], Nicolas Marchand[8], Liqing Hao[9], Annele Virtanen[9], Kalliopi Florou[10], Colin O'Dowd[11], Jurgita Ovadnevaite[11], Urs Baltensperger[1], and André S. H. Prévôt[1]

[1]Laboratory of Atmospheric Chemistry, Paul Scherrer Institute, 5232 Villigen PSI, Switzerland
[2]Department of Chemical Engineering, Carnegie Mellon University, Pittsburgh, USA
[3]TNO, Department of Climate, Air and Sustainability, Utrecht, the Netherlands
[4]Italian National Research Council – Institute of Atmospheric Sciences and Climate, Bologna, Italy
[5]Institute of Environmental Assessment and Water Research (IDAEA), CSIC, 08034 Barcelona, Spain
[6]Institut National de l'Environnement Industriel et des Risques (INERIS), Verneuil-en-Halatte, France
[7]Laboratoire des Sciences du Climat et de l'Environnement (LSCE), Gif-sur-Yvette, France
[8]Aix-Marseille Univ, CNRS, LCE, Marseille, France
[9]Department of Applied Physics, University of Eastern Finland, P.O. Box 1627, FI-70211 Kuopio, Finland
[10]Department of Chemical Engineering, University of Patras, 26500 Patras, Greece
[11]School of Physics, Ryan Institute's Centre for Climate and Air Pollution Studies, and Marine Renewable Energy Ireland, National University of Ireland Galway, University Road, Galway, H91 CF50, Ireland
[*]now at: Institute of Chemical Engineering Sciences, Foundation for Research and Technology Hellas (FORTH/ICE-HT), Patras, Greece

*Correspondence to*: Sebnem Aksoyoglu (sebnem.aksoyoglu@psi.ch), Jianhui Jiang (jianhui.jiang@psi.ch)

**Abstract.** Source apportionment of organic aerosols (OA) is of great importance to better understand the health impact and climate effects of particulate matter air pollution. Air quality models act as potential tools to identify OA components and sources at high spatial and temporal resolution, however, they generally underestimate OA concentrations, and comparisons of their outputs with an extended set of measurements are still rare due to the lack of long-term experimental data. In this study, we addressed such challenges at the European level. Using the regional air quality model Comprehensive Air Quality Model with Extensions (CAMx) and a volatility basis set (VBS) scheme which was optimized based on recent chamber experiments with wood burning and diesel vehicle emissions, and contained more source-specific sets compared to previous studies, we calculated the contribution of OA components and defined their sources over a whole-year period (2011). We modelled separately the primary and secondary OA contributions from old and new diesel and gasoline vehicles, biomass burning (mostly residential wood burning and agricultural waste burning excluding wildfires), other anthropogenic sources (mainly shipping, industry and energy production) and biogenic sources. An important feature of this study is that we evaluated the model results with measurements over a longer period than in the previous studies which strengthens our confidence in our modelled source apportionment results. Comparison against positive matrix factorization (PMF) analyses of aerosol mass



spectrometric measurements at nine European sites suggested that the modified VBS scheme improved the model performance for total OA as well as the OA components, including hydrocarbon-like (HOA), biomass burning (BBOA) and oxygenated components (OOA). By using the modified VBS scheme, the mean bias of OOA was reduced from -1.3 µg m$^{-3}$ to -0.4 µg m$^{-3}$ corresponding to a reduction of mean fractional bias from -45% to -20%. The winter OOA simulation, which was largely

5   underestimated in previous studies, was improved by 29% - 42% among the evaluated sites compared to the default parameterization. Wood burning was the dominant OA source in winter (61%) while biogenic emissions contributed ~55% to OA during summer in Europe on average. In both seasons, the other anthropogenic sources comprised the second largest component (9% in winter and 19% in summer as domain average), while the average contributions of diesel and gasoline vehicles were rather small (~5%) except for the metropolitan areas where the highest contribution reached 31%. The results

10  indicate the need to improve the emission inventory to include currently missing and highly uncertain local emissions, as well as further improvement of VBS parameterization for winter biomass burning. Although this study focused on Europe, it can be applied in any other part of the globe. This study highlights the ability of long-term measurements and source apportionment modelling to validate and improve emission inventories, and identify sources not yet properly included in existing inventories.



## 1 Introduction

Pollution by atmospheric fine particulate matter (PM) exerts significant impacts on human health (Ciarelli et al., 2019; Cohen et al., 2017; Lelieveld et al., 2015; Tuet et al., 2017) and climate (Kanakidou et al., 2005), where organic aerosols (OA) contribute to 20% – 90% (Jimenez et al., 2009; Kanakidou et al., 2005). Unlike the single-component pollutants such as ozone

and sulfur dioxide, organic aerosols are composed of numerous compounds from different sources with distinct physical, chemical and toxicological properties (Hallquist et al., 2009). The situation is even more complicated for secondary organic aerosol (SOA), which is generated from the oxidation of organic gases emitted from a wide range of biogenic and anthropogenic sources and accounts for a dominant fraction of OA (Hodzic et al., 2016; Srivastava et al., 2018). Understanding and identifying the OA sources is therefore important for understanding the implication of aerosols for health and climate and

establishing effective mitigation policies.

Large efforts have been devoted to determine OA sources at different scales, mostly based on receptor modelling. Positive matrix factorization (PMF) analysis is often applied to aerosol mass spectrometer data to classify the measured organic mass spectra into different factors. Commonly retrieved factors include hydrocarbon-like OA (HOA) from traffic emissions, biomass burning (BBOA), cooking (COA), and oxygenated components (OOA) (Crippa et al., 2014). Recent studies with

long-term offline aerosol mass spectrometer (AMS) measurements were able to further split the OOA component into biogenic SOA and anthropogenic SOA according to their seasonal variability (Daellenbach et al., 2016; Daellenbach et al., 2017; Vlachou et al., 2018).

Air quality models (AQMs) provide another approach to quantify OA sources, with great advantages at high temporal and spatial resolution. Various tools have been developed and implemented in AQMs for regional scale source apportionment

of particulate matter, such as the Particulate Source Apportionment Technology (PSAT) for CAMx (Comprehensive Air quality Model with Extensions) (Koo et al., 2009), the tagged species source apportionment (TSSA) (Wang et al., 2009) and integrated source apportionment method (ISAM) (Kwok et al., 2013) for CMAQ (Community Multiscale Air Quality). However, the performance of these tools for OA source apportionment are always limited by substantial underestimation of SOA by the traditional AQMs (Hodzic et al., 2010; Tsigaridis et al., 2014). One of the most important reasons for the

underestimation is the potentially high but unaccounted contribution of non-traditional vapors. To take into consideration these vapors, the volatility basis set (VBS) scheme has been developed and implemented in several air quality models such as CAMx (Ciarelli et al., 2016; Ciarelli et al., 2017a; Koo et al., 2014), CMAQ (Jathar et al., 2017; Koo et al., 2014; Woody et al., 2016), PMCAMx (Lane et al., 2008; Tsimpidi et al., 2010), CHIMERE (Cholakian et al., 2018; Zhang et al., 2013; Zhang et al., 2015), EMEP (Bergström et al., 2012) and WRF-Chem (Ahmadov et al., 2012; Shrivastava et al., 2011). The VBS

scheme classifies the first generation oxidation products of vapors according to their volatility. The evolution of these products with aging through functionalization and fragmentation can also be presented by shifting the volatility of compounds (Donahue et al., 2006). It is widely reported that implementing VBS schemes improves the model performance for SOA (Ciarelli et al., 2016; Fountoukis et al., 2014; Koo et al., 2014; Robinson et al., 2007; Tsimpidi et al., 2010). It is however still a challenge to



use the VBS scheme together with the source apportionment tools in AQMs. To our knowledge, the VBS scheme in CAMx is currently not enabled to be used with the source apportionment tool PSAT (Ramboll, 2018).

A number of studies have used air quality models with VBS to model OA components or sources on regional scales. Bergström et al. (2012) tested the EMEP model with different VBS setups to calculate the contributions of biogenic and

anthropogenic SOA, residential wood combustion and wild fire emissions to OA during 2002 - 2007 over Europe, and evaluated the source apportionment results with observations at four sites with weekly or daily filter measurements of elemental and organic carbon (EC/OC) and one site with hourly AMS measurement. Yttri et al. (2019) used similar EMEP-VBS including reactions of semi volatile organic compounds (SVOC) and intermediate volatility organic compounds (IVOC) to model OA sources at nine rural sites in Europe, and compared the model output with source apportionment of measured EC/OC by

chemical and [14]C tracers. Both studies found that the residential wood burning was largely underestimated, and required further improvement.

Based on recent chamber experimental studies on wood burning (Bruns et al., 2016), Ciarelli et al. (2017a; b) parameterized a hybrid volatility basis set which included a new set for the oxidation from SVOCs and implemented it in the air quality model CAMx to simulate winter OA sources in Europe. The new parameterization significantly improved the model

performance for SOA (Ciarelli et al., 2017a). However, as the biomass burning and biogenic precursors are merged into the same set in the original VBS scheme of CAMx, the biogenic SOA is implicitly taken into account to react with OH in the gas phase, which could lead to overestimated SOA in summer when biogenic emissions are high (Ramboll Environ, 2016).

Skyllakou et al. (2017) extended the PSAT tool in the regional model PMCAMx with VBS to quantify the OA sources in Europe, with major focus on the source-receptor relationship but less attention on the evaluation of OA predictions against

observations. More modelling studies using AQM with VBS to simulate OA sources are available at city scale. Woody et al. (2016) used CMAQ-VBS to model the POA from meat cooking, gasoline and diesel vehicles, biomass burning and other sources in California in May – June in 2010, while the SOA was characterized by formation pathways (including first-product of anthropogenic and biogenic VOCs, first-product of IVOCs, aging reactions of secondary SVOCs, anthropogenic and biogenic VOCs) instead of sources. Jathar et al. (2017) simulated the sources of POA and SOA in southern California with an

updated CMAQ–VBS with special focus on the gasoline and diesel vehicles, and predicted that gasoline vehicles contribute ~35% of the inland OA which is ~13 times more than diesel sources. However, the contribution of gasoline and diesel vehicle emissions to the total OA in Europe, where the vehicle types have high spatial variations, remains unclear. Results from recent aging studies of diesel and gasoline exhaust (Gentner et al., 2016; Platt et al., 2017; Zhao et al., 2017) have not yet been implemented into models either. Despite all the progress, the parameterization of volatility basis sets in AQMs requires further

improvement based on the advance of chamber experimental data, especially for wood burning which was highly underestimated; most modelling studies focus more on the SOA components differentiated by the formation pathways (i.e. from IVOCs, SVOCs), while the sources of SOA are equally important for emission reduction strategies; modelled source apportionment results need further evaluation using measurement data with higher time resolution and longer periods.





In this study, we i) modified the regional air quality model CAMx with the VBS scheme to differentiate primary and secondary organic aerosols from various sources including gasoline vehicles, old and new diesel vehicles, biomass burning (excluding wild fire), other anthropogenic sources and biogenic sources, ii) updated the parameterization of VBS based on recent smog chamber experimental data for residential wood burning and diesel vehicles, with separate sets and parameterization for aging of secondary condensable gases from biomass burning and biogenic sources, iii) conducted a whole year simulation in 2011 to calculate the OA concentrations from different primary and secondary sources in Europe, and iv) evaluated the model performance on OA source apportionment  by comparing the model results with the PMF analyses of hourly observational data covering nine Aerodyne aerosol chemical speciation monitor (ACSM) /Aerodyne aerosol mass spectrometer (AMS) stations in Europe with measuring periods ranging from one month to one year.

## 2 Method

### 2.1 Air quality model CAMx

The air quality model CAMx version 6.3 (Ramboll Environ, 2016) was used to simulate OA for the full year 2011. The model domain covers Europe (15ºW - 35ºE, 35ºN - 70ºN), with a spatial resolution of $0.25º \times 0.125º$ and 14 terrain-following vertical layers ranging from ~20 m above ground level (1$^{st}$ layer) going up to 460 hPa. The Carbon Bond 6 Revision 2 (CB6r2) gas-phase mechanism (Hildebrandt Ruiz and Yarwood, 2013) was selected in this study. The ISORROPIA thermodynamic model (Nenes et al., 1998) was used to simulate gas-aerosol partitioning of inorganic aerosols. For organic aerosols, a modified VBS module based on the 1.5-D VBS scheme (Koo et al., 2014) was used to model the formation and evolution of OA. The meteorological parameters were produced with the Weather Research and Forecasting model (WRF, version 3.7.1; Skamarock et al., 2008), based on the 6-h European Centre for Medium–Range Weather Forecasts (ECMWF) reanalysis global data with resolution of $0.72º \times 0.72º$ (Dee et al., 2011). The initial and boundary conditions for the concentrations of chemical species were obtained from the global model MOZART-4/GEOS-5 (Horowitz et al., 2003). The Total Ozone Mapping Spectrometer (TOMS) data by the National 25 Aeronautics and Space Administration (ftp://toms.gsfc.nasa.gov/pub/omi/data/) was adopted for the input of ozone column densities, and the photolysis rates were calculated by the Tropospheric Ultraviolet and Visible (TUV) Radiation Model version 4.8 (NCAR, 2011). A more detailed description about the inputs of meteorology, photolysis, initial and boundary conditions is found in Jiang et al. (2019). The simulation period was between 1 January 2011 and 31 December 2011 with the first two weeks being used as spin-up. We performed the simulations using both the standard (BASE) and modified (NEW) parameterization in the VBS module.



## 2.2 VBS parameterization

### 2.2.1 Extended volatility basis sets

The default VBS scheme of CAMx version 6.3 includes five basis sets to describe the oxidation process of OA: three sets for freshly emitted OA from biomass burning (PFP, Particle Fire Primary), cooking (PCP, Particle Cooking Primary) and other

anthropogenic (PAP, Particle Anthropogenic Primary), and two basis sets for chemically aged oxygenated OA from anthropogenic (PAS, Particle Anthropogenic Secondary) and biogenic (PBS, Particle Biogenic Secondary) emissions. For the set PAS, the OA generated from gasoline vehicles (GV), diesel vehicles (DV) and other anthropogenic activities (OP) are parameterized according to volatility distributions, yields and vaporization enthalpies reported in the literature, and then merged to PAS. A similar treatment is adopted for secondary OA from biomass burning, cooking and biogenic sources, which

are merged together as PBS.

We have modified the default VBS to consider additional sources. As the first step to separate the modelled OA components, the standard 5 basis sets were split into 11 basis sets including primary and secondary OA from 5 sources, i.e., new diesel vehicles (DN, >= Euro 4) equipped with diesel particle filter (DPF), old diesel vehicles (DO, < Euro 4) without DPF, gasoline vehicles (GV), biomass burning (BB), and other anthropogenic sources (OthA), as well as SOA from biogenic

sources (BIO). The schematic diagram of the VBS with the modified basis sets is shown in Fig. 1. Due to lack of emission data, OA from cooking emissions was excluded in this study. Instead of merging OA from different sources as done in the default CAMx-VBS, we added the species in the 11 basis sets to the species list of model output to distinguish the OA sources. The new VBS scheme was then tested with the standard parameterization of CAMx version 6.3 (BASE) and a modified parameterization (NEW) based on recent experimental findings.

### 2.2.2 Standard parameterization: BASE

The BASE parameterization contains the default parameters of the CAMx (version 6.3) VBS module. The same parameterization (volatility distribution, yields, reaction rates of VOC/IVOC precursors and primary/secondary condensable gases) of the standard set PAS were used for the secondary OA from DO, DN, GV, OthA, as well as the PAP for the primary OA from DO, DN, GV, OthA. The primary and secondary BB and BIO followed the same parameterization as PFP and PBS,

respectively. The aging of biogenic SOA was disabled in the standard parameterization as it led to significant over-prediction of OA in rural areas in previous modelling studies (Ramboll Environ, 2016). As the biomass burning and biogenic SOA belong to the same PBS set and use the same parameterization in the default version, the aging of biomass burning SOA was also disabled in BASE.

### 2.2.3 Modified parameterization: NEW

In the NEW case, we used a modified parameterization based on smog chamber experimental studies. The major changes were made for diesel vehicles and biomass burning. Diesel vehicles constitute nearly half of the total passenger car registrations in





Europe (ACEA, 2017). Diesel vehicle emissions were traditionally considered more efficient at generating SOA than gasoline exhaust (Gentner et al., 2012). However, the vehicles equipped with diesel particle filter (DPF) were found to effectively reduce SOA production (Gentner et al., 2016; Gordon et al., 2013; Platt et al., 2017). In Europe, DPFs have been implemented in some diesel vehicles since Euro 4 in 2005 and have been required for all diesel vehicles since Euro 5 in 2009. Due to a large

share of diesel vehicles equipped with DPF, we set the SOA yield for the basis set DN (diesel vehicle – new) to zero. Residential biomass burning is among the largest winter OA sources in Europe (Crippa et al., 2014; Lanz et al., 2010). Models generally underestimate OA from biomass burning (Hodzic et al., 2010). The NEW parameterization enables the oxidation of secondary gases from biomass burning (see BB in Fig. 1) with a reaction rate of $4\times10^{-11}$ cm$^3$ molec$^{-1}$ s$^{-1}$ according to Ciarelli et al. (2017a; 2017b). For other basis sets, the default parameters of CAMx v6.3 were used.

**2.3 Emissions**

The anthropogenic emissions were based on the high-resolution European emission inventory TNO-MACC (Monitoring Atmospheric Composition and Climate)-III, which is an extension of the TNO-MACC-II (Kuenen et al., 2014). The annual emissions of non-methane volatile organic compounds (NMVOCs), $SO_2$, $NO_x$, CO, $NH_3$, $PM_{10}$ and $PM_{2.5}$ were hourly distributed using the TNO temporal variation profiles. The fine and coarse particulate matter emissions ($PM_{2.5}$ and $PM_{10}$,

respectively) were split into POA, elemental carbon (EC), sodium (Na), particulate sulfate ($PSO_4$), and other primary particles in the fine and coarse size fractions according to the TNO PM-splitting profile. The NMVOC speciation was performed using the approach of Passant (2002) to generate emissions of 20 NMVOC species including toluene, xylene and benzene. The TNO-MACC-III emission inventory includes anthropogenic emissions from 9 SNAP (Selected Nomenclature for Air Pollution) source categories: energy industries (SNAP 1), residential and other non-industrial combustion (SNAP 2), industry combustion

and processes (sum of SNAP 3 and SNAP 4), extraction and distribution of fossil fuels (SNAP 5), product use (SNAP 6), road transport (SNAP 7), non-road transport and other mobile sources (SNAP 8), waste treatment (SNAP 9) and agriculture (SNAP 10). Additional SNAP 7 emissions were provided by TNO, which include more detailed classification of gasoline and diesel vehicles (before Euro 4 and for EURO 4 and higher emission standards), liquefied petroleum gas vehicles, vehicles evaporation and brake wear. To match the requirement of the modified VBS module, we reclassified the diesel vehicles before and after

(including) Euro 4 as OLD and NEW diesel vehicles (DO and DN), respectively. All the other emissions from road traffic such as vehicular evaporation and brake wear were attributed to the "other anthropogenic (OthA)" due to the lack of information to conduct a specific parameterization. The POA from residential combustion (SNAP2) and agriculture (SNAP10, mainly from on-field burning of stubble and straw) were summed up to represent POA from biomass burning as justified in Ciarelli et al. (2017a). The wildfires are not included in the biomass burning.

The intermediate-volatility and semi-volatile organic compounds (IVOCs and SVOCs) were considered as important precursors of SOA (Jathar et al., 2011; Jathar et al., 2014), but they are generally absent in current emission inventories. Here we estimated the IVOC emissions from different sources based on literature (Table S1). The IVOCs from gasoline and diesel vehicles were calculated as 25% and 20% of NMVOC emissions from gasoline and diesel vehicles, respectively, according to



the gas-phase carbon-balance analysis of Jathar et al. (2014). IVOC emissions from biomass burning were estimated as 4.5 times of POA emissions based on Ciarelli et al. (2017a). For other anthropogenic sources, the IVOC emissions were calculated as 1.5 times of POA as proposed by Robinson et al. (2007), which was widely adopted by modelling studies (Jathar et al., 2017; Woody et al., 2016). The VBS scheme assumes that a certain fraction of POA can evaporate and be distributed in the

5 semi-volatile range (saturation concentration between 0.1 to 1000 µg m$^{-3}$), while most of the emission inventories only include POA in the particle phase. According to the partitioning theory (Donahue et al., 2006), the ratio between gas and particle phase in the semi-volatile range is roughly 3, and therefore many modelling studies increase the POA emissions by a factor of 3 to compensate for the missing SVOCs (Ciarelli et al., 2017a; Shrivastava et al., 2011; Tsimpidi et al., 2010). This approach agrees well with the emission study in Europe which shows that the revised residential wood combustion emissions accounting for

the semi-volatile components are higher than those in the previous inventory by a factor of 2-3 on average (Denier van der Gon et al., 2015). However, Denier van der Gon et al. (2015) also pointed out that this factor presents substantial inter-country variability due to different combustion type, fuel parameters and operation conditions, indicating a potential over- or underestimation for a specific area by using the factor of 3 in the whole domain. To investigate the role of SVOC, we adopted the approach to increase POA emissions by a factor of 3 in NEW, while keeping the POA unchanged in BASE.

Biogenic emissions (isoprene, monoterpenes, sesquiterpenes, soil NO) were estimated by the PSI model developed at the Laboratory of Atmospheric Chemistry at the Paul Scherrer Institute (Andreani-Aksoyoglu and Keller, 1995) and further improved by Oderbolz et al. (2013) and Jiang et al. (2019). A comparison study with the widely used biogenic model MEGAN (Model of Emissions of Gases and Aerosols from Nature) version 2.1 indicated that the PSI model produces higher monoterpene emissions in Europe than MEGAN, and leads to a better performance of CAMx for OA (Jiang et al., 2019).

**2.4 Model evaluation**

As the performance of CAMx strongly depends on the quality of the meteorological inputs, we first evaluated the meteorological parameters (surface temperature, wind direction, wind speed, precipitation) modelled by WRF as described in Jiang et al. (2019) using observations obtained from the UK Met Office Integrated Data Archive System (MIDAS) Land Surface Stations database (Meteorological Office, 2013). It covers ~1000 stations in Europe and provides observations at 3-

25 hour time resolution. The model performance criteria used for meteorological parameters (Emery, 2001)( are shown in Table S2. The general model performance for the main gas-phase species (i.e. $O_3$, $SO_2$, $NO_x$, CO) and fine particulate matter $PM_{2.5}$ was also evaluated using the hourly measurements extracted from the European Environment Agency database, AirBase v7 (Mol and Leeuw, 2005). The statistical analysis was conducted by means of mean bias (MB), mean gross error (MGE), root-mean-square error (RMSE), mean fractional bias (MFB), mean fractional error (MFE), index of agreement (IOA) and

30 correlation coefficient ($r$) between measured and modelled results. For ozone, only measurements at the background-rural stations were used in the model evaluation to reduce the possible uncertainties caused by the model resolution.

The model performance for OA from different sources in Europe was evaluated using the OA measurements and source apportionment studies using positive matrix factorization (PMF) analysis, covering nine ACSM/AMS stations over Europe:



Zurich (Canonaco et al., 2013), Mace Head (Ovadnevaite et al., 2014; Schmale et al., 2017), Montsec (Ripoll et al., 2015), Bologna (Paglione et al., 2019) and San Pietro Capofiume (SPC) (Gilardoni et al., 2014), Paris SIRTA (Site Instrumental de Recherche par Télédétection Atmosphérique) (Petit et al., 2015), Marseille (Bozzetti et al., 2017), Finokalia (as continuation of Hildebrandt et al. (2010)), and SMEAR (Station for Measuring Forest Ecosystem–Aerosol Relations) II Hyytiälä

(Kortelainen et al., 2017). The types and locations as well as the measurement periods of stations are displayed in Table 1 and Fig. S1. The meteorological measurements for Paris, Marseille and Finokalia were obtained from the UK Met Office Integrated Data Archive System (MIDAS) Land Surface Stations database (Meteorological Office, 2013), for Zurich from the automatic monitoring network of MeteoSwiss (ANETZ), and for the other stations the meteorological data measured at or near the stations were provided by the measurement groups.

## 3. Results and discussion

### 3.1 Model evaluation

#### 3.1.1 Meteorological parameters and major air pollutants

Modelled and measured meteorological parameters showed good agreement (Table S2). Most of the parameters fulfill the criteria for meteorological model performance (Emery, 2001) except surface temperature in winter (underestimated by

15 about -1°C) and the wind direction in summer. Most of the stations where surface temperature underestimation occurs are located in the Alpine regions and the Baltic coast, and the high bias of wind direction mostly occurs in the Mediterranean region, with a limited influence on the whole domain.

The model performance for the major air pollutants such as $O_3$, $NO_2$, $SO_2$ and $PM_{2.5}$ is presented in Table S3. As the modified parameterization for OA has a negligible impact on the gas species, the statistical results of the gaseous pollutants

are displayed only for the NEW case while the results for $PM_{2.5}$ are shown for both the BASE and NEW cases. For both winter and summer, the recommended model performance criteria and goals (Table S4, based on Boylan and Russell (2006) and EPA (2007)) were met for $PM_{2.5}$ (both BASE and NEW) and ozone. The $NO_2$ concentrations were underestimated (MFB: -43% in winter, -36% in summer), as reported by other European modelling studies as well (Bessagnet et al., 2016; Ciarelli et al., 2016; Knote et al., 2011; Oikonomakis et al., 2018) and might be related to underestimated emissions. The $SO_2$ concentrations were

over-predicted with a mean bias of 6.7 ppb and 3.9 ppb in winter and summer, respectively. The highest overestimation occurred at sites with high $SO_2$ emissions, e.g. harbors and Eastern Europe. One reason could be the accumulation in the first layer, since the $SO_2$ emissions were all injected to the first layer of the model. The high uncertainties in ship emissions could be another reason for the overestimated $SO_2$ concentrations (Aksoyoglu et al., 2016). Overall, the model performance for the major air pollutants was comparable to the results of other modelling studies in Europe performed in the framework of the

EURODELTA III exercise (Bessagnet et al., 2016).



### 3.1.2 Total organic aerosol

We evaluated the model results for OA and its components calculated both with the standard parameterization (BASE) and with the modified VBS module (NEW) using the measurements available at 9 ACSM/AMS stations (Table 2). There were two main differences between the two cases: i) the BASE case used the standard POA emissions while in the NEW case, POA

emissions were three times higher, ii) the further aging of secondary biomass burning gases was enabled in the NEW case. In the NEW case, OA concentrations are higher, leading to a decreased mean bias (by 0.5 to 7.7 μg m$^{-3}$) between modelled and measured OA at most sites except for San Pietro Capofiume and SMEAR II. As a consequence of major changes in the parameterization related to biomass burning and road traffic, the improvement in model performance is more significant at sites close to urban areas where the contribution of anthropogenic sources is relatively higher e.g. Marseille (urban background

site, located in a park near the city center), Paris SIRTA (suburban background site, located 25 km Southwest of Paris city center) and Bologna (urban background site, located at the northwestern edge of the city, surrounded by distributed industrial and agricultural activities and major highways), with a reduction in MFE from 37% in Bologna to 47% in Paris SIRTA. For another urban background site Zurich, the NEW parameterization led to a decrease in the MB (by 0.5 μg m$^{-3}$), MFB (by 20%) and MFE (by 8%), while RMSE and MGE slightly increased by 0.4 μg m$^{-3}$ and 0.2 μg m$^{-3}$, respectively. For the rural/remote

sites, the CAMx-NEW reduced the MFE by 3% to 12% for Mace Head, Montsec and Finokalia, but it led to an overestimation at SPC and SMEAR II.

In order to further investigate the reasons for different model performance at different sites, the temporal variations of modelled OA for CAMx-BASE and CAMx-NEW were compared with the measurements (Fig. 2, Fig. S2). We selected four stations to represent different site types, i.e. Zurich as an urban (background) station, Paris SIRTA as a suburban station, SPC

as a rural station, and Montsec as a remote station. Among the four sites, measurements in Zurich cover almost the whole year of 2011, while measurements at other stations took place mostly in autumn and winter periods. The modelled OA concentrations in Zurich matched the measurements quite well except in winter, when the model underestimated OA with the BASE parameterization (Fig 2a). In addition, there was a significant underestimation of temperature, which was expected to increase the modelled OA via enhanced condensation, indicating that the predicted OA concentrations would have been even

lower if the modelled temperature had been higher. Including the reactions of SVOCs and further aging of secondary condensable gases for biomass burning sources in CAMx–NEW, the model performance in winter was effectively improved by increasing the OA concentrations by ~100% compared to CAMx–BASE during February and March. Nevertheless, CAMx–NEW led to an overestimation of OA in September and November, which influenced the overall performance of NEW. The two peaks in September and October were largely due to the SOA from biogenic emissions, according to our previous work

comparing effects of two different biogenic emission models on OA (Jiang et al., 2019). Paris SIRTA is among the sites with best improvement of the predicted OA. In spite of the underestimation, the modelled OA reproduced the temporal evolution of the major peaks during the measurement period and the OA by CAMx–NEW is about ~1.4 times higher than CAMx–BASE





(Fig. 2b). The highest peak during 19–24 November was related to an air mass from south-southeast (SSE) with a recirculation over northern France, which is dominated by wood burning source (Petit et al., 2015).

At the rural site, SPC, a distinction can be seen before and after 23 November (Fig. 2c). Both CAMx–BASE and CAMx–NEW overpredicted the OA before 23 November. Located in the Po Valley, OA in SPC is significantly influenced by fog

scavenging processes during autumn when the high relative humidity and low temperature lead to frequent fog events (Gilardoni et al., 2014). Due to the coarse resolution and special geographical location of SPC, the meteorological model failed to reproduce the extremely high relative humidity during the fog events, leading to an overestimation of OA as particle activation into fog droplets, and eventually wet scavenging, are not adequately captured by the model. A better agreement with the observed OA was found for CAMx–NEW than CAMx–BASE after 23 November, when fog events were shorter.

For comparison with the observations at the high-altitude (~1570 m above sea level) station Montsec, we used the model results from the 5[th] vertical layer. Generally, air quality model results are poorly reproduced at high-altitude sites due to domain resolution. Although the NEW parameterization increased the OA by 35%, the OA concentrations were still largely underestimated during July to September (Fig. 2d). The temporal variation of OA in Montsec is mainly influenced by the special meteorological conditions and the local planetary boundary layer (PBL) height. Especially in summer, the higher

temperature and solar radiation enhances the growth of PBL and transport of OA to high elevation (Ripoll et al., 2015).

## 3.2 Comparison against PMF results

The modelled OA components were evaluated using AMS/ACSM measurements analyzed with PMF at different stations (Table 2). The CAMx–NEW led to a better agreement between the modelled and measured primary OA at most sites, except in Zurich (where HOA and BBOA tended to be overestimated by CAMx–NEW and underestimated by CAMx–BASE). In

general, the NEW parameterization improved the underestimation of OOA by reducing the mean bias by 0.3 $\mu$g m$^{-3}$ (SMEAR II) to 1.7 $\mu$g m$^{-3}$ (Bologna), while increasing the mean bias at Zurich and SPC where OOA concentrations were overestimated by both CAMx–BASE and CAMx–NEW. To better understand the different model performance, the modelled and measured OA components were compared separately for the four different seasons (Fig. 3). The overestimation of OOA in autumn at SPC was largely due to the absence of fog droplet activation in the model as discussed in Section 3.1.2. In spite of the

overestimation for the whole period, the winter OOA at Zurich was still underestimated by 73% (BASE) and 64% (NEW). The underestimation of winter OOA was also found at all the other studied sites (Marseille, Bologna and Paris SIRTA), while the summer OOA showed good agreement with the PMF results (Fig. 4a and 4d), indicating a possible underestimation of SOA from biomass burning (Gilardoni et al., 2014; Gilardoni et al., 2016; Paglione et al., 2019; Qi et al., 2019). The NEW parameterization improved the OA modelling by increasing winter OOA by 29.3% (SIRTA) to 41.7% (Bologna). Compared

to the modelled BBOA (Fig. 4b and 4e) and HOA (Fig. 4c and 4f) which were improved considerably by using the NEW parameterization, the SOA modelling still needs to be improved.



A more detailed comparison between the temporal variations of the OA components obtained by CAMx-NEW and PMF is presented in Fig. 5 and Fig. S3. As an example of an urban background site, at Zurich, the contributions of POA and SOA from different sources generally agree well with the PMF results. The OOA from biogenic sources (OOA-BIO) begins to increase from April with increasing temperature and biomass density (Fig. 5a), similar to the semi-volatile oxygenated organic
aerosol (SV-OOA) by PMF (Fig. 5b), which is mainly produced from biogenic precursor gases in summer (Canonaco et al., 2015). However, for the winter period, the modelled contribution of BBOA is much higher than the PMF results, whereas the contribution of the total OOA is under-predicted. This may be partly linked to uncertainties in PMF analysis for discriminating BBOA and OOA (Crippa et al., 2013; Petit et al., 2014). At some sites (e.g., Paris SIRTA), biomass burning emissions could also have substantial contributions to the HOA and/or COA-like PMF factors (Petit et al., 2014; O. Favez, personal
communication). It might also indicate the need for further improvement on the model parameterization of the biomass burning sector. For the remote site Montsec, despite the underestimated total OA (Fig. 2d, largely due to meteorological conditions), the contributions of the OA components are close to the PMF results (Fig. 5c-d). BBOA was not identified in the PMF study during the investigated period, while the modelled results show an average mass fraction of 13%. SOA dominated the OA fraction for both modelled and PMF results, with a ratio of 87% and 78%, respectively. OOA from biogenic sources constitutes
the largest OA fraction (~53%) according to the modelled results, which is in agreement with the previous findings in the Mediterranean forested area of Montseny (Ripoll et al., 2015).

The model performance in reproducing the diurnal variations of the OA components varies with sites. Figure 6 shows the diurnal variations of HOA, BBOA and OOA by CAMx-NEW and PMF in winter. The simulation of primary organic aerosols (both HOA and BBOA) at site scale largely depend on the anthropogenic emission inputs. The modelled HOA shows
dual peaks in the morning and evening rush hours for all sites as a consequence of the same diurnal variation factors in the emission model, while similar dual peaks only occur in Marseille and Bologna for the PMF HOA. The PMF HOA peaks are less pronounced in the urban background site Zurich and the suburban site SIRTA, where the magnitude of HOA is also lower compared to the other two urban stations. Similar to HOA, the modelled BBOA generally showed dual peaks in the morning and evening cooking and heating time. However, the diurnal pattern of PMF BBOA showed more variability over the sites.
The largest difference between modelled and PMF BBOA occurred in Marseille, where the peaks of PMF BBOA were generally later than the modelled results and much higher at night. The high PMF BBOA during night mainly comes from the BBOA transported from the valleys near Marseille by the night land breeze, whereas the local meteorology was poorly reproduced with largely underestimated wind speed at Marseille as a common problem at coastal sites for meteorology modelling (Fig. S2b). Another important reason for the underestimated BBOA is that green waste combustion and agricultural
fires comprise a large fraction of the BBOA in Marseille in February (Bozzetti et al., 2017), however, this part of BBOA was not modelled due to lack of emission data. To improve the model performance of the primary OA, it is necessary to further improve the emission inputs by including more site-specific emission sources, as well as updated diurnal variation profiles to get better agreement with the observations. In spite of a general underestimation for the modelled OOA in winter, both the CAMx–NEW and PMF OOA have rather flat diurnal patterns (Fig. 6). A better agreement between modelled and PMF results


is found at rural/remote sites during spring to autumn (Fig. 7), especially at SMEAR II, Finokalia and Montsec, where OOA from biogenic sources contribute more than 90% of the total OA according to the PMF results. The PMF OOA at San Pietro Capofiume shows a peak during the day when the liquid water content was low (Gilardoni et al., 2014), and the peak is higher during the less foggy period after 23 November (Fig. 7e) than during the strong foggy period (Fig. 7d). However, the modelled

OOA is flat as the fog scavenging effect is poorly reproduced.

### 3.3 Spatial and seasonal variation of OA components and sources

Due to the better performance of CAMx–NEW in predicting the OA components, the spatial and temporal variations of OA components and sources were investigated using the CAMx–NEW parameterization. The contributions of gasoline and diesel vehicles, biomass burning, other anthropogenic sources (including shipping, energy sector and industry), and biogenic

sources in winter and summer are displayed in Fig. 8. Country-scale relative contribution of each source to the total OA can be found in Table S5. In winter, the largest contribution over the whole domain comes from biomass burning from residential and agriculture activities (Fig. 8a), and the biomass burning POA is the dominant component, with an average fraction of 43% and the highest value reached in Slovenia (19.9 μg m$^{-3}$, Fig. S4). The second largest OA component is the biogenic SOA (25%), followed by the biomass burning SOA (19%). The high contribution of biogenic SOA in winter is mainly found in

Sweden, Ireland, and Spain (Table S5). Ireland and Spain have high emissions of the biogenic SOA precursor monoterpenes due to a high coverage of Norway spruce trees, the major emitters of monoterpenes, as well as comparatively high temperature. Our biogenic emission model assumes that grids with snow coverage more than 50% have zero monoterpene emissions in winter (Jiang et al., 2019). Therefore, there are still considerable monoterpene emissions in winter in areas with low snow coverage such as Ireland and Spain. The high biogenic SOA fraction in Sweden is mostly due to low SOA from anthropogenic

activities.

Significantly higher contributions of SOA components were found in summer, when the biogenic SOA contributed up to 55% of total OA in Europe. The other anthropogenic sources (OthA) are the second largest contributor except for a small region in the Balkans where biomass burning shows higher contribution. SOA from OthA was higher than POA with an average contribution to total OA of 13%. Unlike the dominant role of biomass burning POA in winter, the SOA from biomass

burning is higher than POA in summer. The model results generally agree with previous studies for Europe, i.e. the biogenic emissions are the dominant OA source in summer, and the residential wood burning are the most important winter OA source (Bergström et al., 2012; Skyllakou et al., 2017). However, the relative contribution of biogenic sources in this study are higher than in other studies especially in winter with an average relative contribution of 25%, while the value is less than 5% in Skyllakou et al. (2017). Jiang et al. (2019) showed that the biogenic emission model we used produces higher monoterpenes

than the widely used MEGAN model, which partly explains the higher contribution of the biogenic sources to the total OA. As the biogenic emissions are associated with high uncertainty while the measurements of biogenic volatile organic compound





emissions are sparse, further studies are still needed to validate biogenic emissions. Meanwhile, the lack of cooking and wildfire emissions might also lead to an increased fraction of biogenic OA in this study.

The contributions of gasoline and diesel vehicles to total OA were rather small compared to the other sources (Fig. 8), with an average fraction of ~5% (3.9% in winter and 6.3% in summer for the sum of POA and SOA). However, the contribution is still high in metropolitan areas like Paris and Milan with a maximum value up to 31% in summer. Although the magnitude of the road-traffic contribution to OA was similar for these "hotspots", major components were found to be different. For Paris, POA from diesel vehicles was identified as the major contributor (Fig. 9b and 9f) as a result of the high share of diesel vehicles in France (ACEA, 2017), while SOA from gasoline vehicles contributed most in the Italian cities including Milan, Rome and Naples (Fig 9c and 9g). The contribution of diesel vehicles to SOA (Fig. 9d and 9f) was much lower than the gasoline vehicles (Fig. 9c and 9g). Compared with recent studies in southern California where the gasoline and diesel vehicles contributed ~35% and ~2.6% to the total OA (Jathar et al., 2017), the highest contribution of gasoline vehicles to the total OA in Europe (20% in Naples, summer) was lower while the contribution of diesel vehicles (24% in Paris, summer) is much higher, due to the distinct vehicle mix in the US and Europe.

## 3.4 Regional OA sources

In order to understand the regional variations of OA components and sources, we divided the model domain into 8 sub-regions: the Iberian Peninsula (IP), the Mediterranean (MD), Po Valley (PV), eastern Europe (EE), central Europe (CE), Benelux (BX), Ireland and Great Britain (IG) and Scandinavia (SC). The regional division and average fractions of the POA and SOA components for each region are shown in Table 3 and Fig. 10. In winter, OA in most regions is dominated by biomass burning POA except for Ireland and Great Britain and the Iberian Peninsula, where the biogenic SOA contributes most with a fraction of 66% and 45%, respectively. Both of these two regions have relatively high monoterpene emissions as explained in Section 3.2. Meanwhile, the high biogenic SOA fraction in Ireland and Great Britain could also come from underestimated contribution of biomass burning, in which the peat combustion covers a considerable portion (Lin et al., 2018) but is not considered in the emission inventory. The second largest POA source in all regions is the other anthropogenic sources, with a similar contribution to total OA (4.9% – 9.6%). The Po Valley and Benelux regions feature a comparatively higher contribution of POA from diesel vehicles (4.5% for PV, 4.9% for BX). For the winter SOA, the Mediterranean, Eastern Europe and Po Valley are dominated by biomass burning (20% – 29%), while the other regions are dominated by biogenic SOA (30% in CE– 66% in BX). The summer OA is dominated by SOA for all the regions due to a considerable contribution of biogenic SOA. Scandinavia shows the highest fraction of biogenic SOA reaching up to 77%, followed by Ireland and Great Britain (70%) and the Iberian Peninsula (55%). In summer, a significant increase is seen in the contribution of gasoline-SOA compared to winter, especially in Central and Southern Europe. The Po Valley has the highest fraction of gasoline-SOA (9.4%), followed by the Mediterranean (7.3%) and central Europe (4.0%). The contributions of the POA components show more regional variation. In the coastal regions e.g. the Mediterranean, Ireland and Great Britain, Benelux and Scandinavia, POA–OthA has





the highest contribution, while in the other regions POA–BB contributes most (although with much lower values than in winter).

Among all the anthropogenic sources, OthA contributes from 9% (EE) to 25% (IG) in winter and from 33% (EE) to 64% (IG) to the anthropogenic OA (excluding the biogenic OA) in summer. This component mostly comes from the non-road
transport, energy sector, industrial production and processes, solvent use, waste treatment, extraction of fossil fuels, as well as brake wear. The CAMx–VBS is not able to further separate the OA sources from OthA. To give a general view about the contribution of specific sources in OthA, the contributions of $PM_{2.5}$ (as a proxy of POA) and NMVOC (as a proxy of gaseous OA precursor) emissions from specific sources are shown in Fig. S5. The highest ratio of $PM_{2.5}$ emissions from OthA reaches 82% in the Iberian Peninsula due to high contribution of non-road transport (shipping), and the highest fraction of NMVOC
emissions from OthA (major proxy of the gaseous precursors from OthA) reaches 86% due to the solvent use. However, compared to the widely studied OA sources such as biomass burning and road traffic, knowledge about these sources is still quite limited.

## 4. Conclusions

This modelling study was conducted to identify the sources of organic aerosol (OA) components in Europe using the air
quality model CAMx for whole year of 2011. In order to improve the model performance for the organic aerosol which is generally underestimated by air quality models, we updated the VBS parameterization based on the recent findings in chamber experiments with biomass burning and diesel vehicles (CAMx-NEW). A more source specific VBS scheme compared with previous studies was used to calculate separately the OA contributions from old and new diesel and gasoline vehicles, biomass burning (residential wood burning and agricultural combustion), other anthropogenic sources (mainly shipping, industry and
energy production) and biogenic sources in Europe. We modified the basis sets and emissions in order to be able to identify OA sources such as gasoline vehicles (GV), old diesel vehicles before Euro 4 emission standards (DO), new diesel vehicles with Euro 4 and higher (DN), biomass burning (BB), biogenic sources (BIO) and other anthropogenic sources (OthA). CAMx-NEW enhances the role of semi-volatile organic compounds (SVOC) and enables the further aging of secondary condensable gases from biomass burning, which significantly improved the model performance for the total OA.

Another important outcome of this study is the evaluation the model results with measurements over a longer period than in other studies which strengthened our confidence on our modelled source apportionment. The model evaluation using measurements at 9 ACSM/AMS stations in Europe showed that the CAMx-NEW reduced the mean fractional error (MFE) between the modelled and measured OA by 3% – 47% compared to the standard VBS parameterization (CAMx-BASE). The model performance to reproduce the OA components (HOA, BBOA and OOA) was also improved by using CAMx-NEW.
The MFE between modelled and PMF HOA was 24% – 72% lower with CAMx-NEW than with CAMx-BASE, and the MFE of BBOA was reduced by 23% – 47% with CAMx-NEW. The general underestimation of OOA by models was also improved with reduced mean bias of 0.3 – 1.7 μg m⁻³ by the modified VBS scheme.





The model results of CAMx–NEW suggested that biomass burning and biogenic sources are the major sources of OA in Europe in winter and summer, respectively. The highest contribution from biomass burning to OA was predicted in the Po Valley and Eastern European regions with > 70% in winter. The SOA from biogenic sources was calculated to cover more than 50% of summer OA on average (highest with 77% in Scandinavia). Over Ireland and Great Britain, the contribution of
SOA from biogenic sources was more than 50% even in winter, possibly due to underestimated biomass burning emissions in that area. The Other Anthropogenic component excluding biomass burning and road traffic (incl. shipping, energy sector, industry, *etc*) was identified as another group of important OA sources with an average contribution of 9% and 19% in winter and summer respectively, and the highest contribution to total OA was predicted to be 29% in the Mediterranean in summer. The contribution of road traffic (gasoline + diesel vehicles) was rather small on average (~5%) but was higher in metropolitan
areas. The highest contribution of gasoline and diesel vehicles to total OA was found in the Po Valley region with a value of 16% in summer.

The results of this study provide information about the OA source apportionment with a regional perspective, which complements the current measurements and PMF analysis to understand regional differences of OA sources in Europe, and to identify needs for future studies. The modelling of OA source apportionment needs further evaluation based on measurements
with wide spatial coverage, especially using advanced measuring techniques with improved capability of SOA identification. The considerable contribution of OA from shipping, industry and energy sectors highlights the importance of more experimental and model studies on such sources to provide explicit parameters (yields, volatility distributions, SVOC emissions, etc.) which are currently estimated by default parameters in most of the air quality models. The emission inventory remains to be improved for the highly uncertain cooking and wildfire emissions, as well as to include more site-specific
emission sources, to further improve the model performance for predicting the OA sources.

*Data availability.* The data of this study will be available online before publication on ACP.

*Author Contribution.* JJ and SA conceived the study. JJ carried out the model modification, simulation and data analysis. GC
contributed to model setup and development of the source apportionment code. HACDG provided the anthropogenic emissions. IEH, FC, SG, MP, MCM, OF, YZ, NM, LH, AV, KF, COD and JO provided the measurement data and contributed to data interpretation. SA, ASHP and UB supervised the entire work development. The manuscript was prepared by JJ. All authors discussed and contributed to the final paper.

*Competing interests.* The authors declare that they have no conflict of interest.

*Acknowledgements.* We would like to thank the European Centre for Medium-Range Weather Forecasts (ECMWF) for the access to the meteorological data, the European Environmental Agency (EEA) for the air quality data, the National Aeronautics and Space Administration (NASA) and its data-contributing agencies (NCAR, UCAR) for the TOMS and MODIS data, the





global air quality model data and the TUV model. We acknowledge the continuous support of CAMx by RAMBOLL. Simulation of WRF and CAMx models were performed at the Swiss National Supercomputing Centre (CSCS). EPA-Ireland (AEROSOURCE, 2016-CCRP-MS-31) is acknowledged, as well as Emilia-Romagna Region "Supersito" Project (DRG 428/10; DGR 1971/2013), EGAR group from IDAEA-CSIC (special mention to Anna Ripoll and Andrés Alastuey) and

Generalitat de Catalunya (AGAUR 2017 SGR41). M.C. Minguillón acknowledges the Ramón y Cajal fellowship awarded by the Spanish Ministry of Economy, Industry and Competitiveness. SIRTA measurements have been conducted in the frame of the EU-FP7 ACTRIS project (grant agreement no. 262254). We also acknowledge the support by the COST Action CA16109 Chemical On-Line cOmpoSition and Source Apportionment of fine aerosoL (COLOSSAL). This study was financially supported by the Swiss Federal Office of Environment (FOEN).

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


**Table 1.** Coordinates, observation periods and PMF related information of stations.

| Station | Latitude | Longitude | Type | Monitor [a] | Observation period | OA components [b] of the PMF data |
|---|---|---|---|---|---|---|
| Bologna | 44.53º N | 11.35º E | Urban | AMS | 01.11.2011 – 06.12.2011 | HOA, BBOA, LV-OOA, SV-OOA |
| Finokalia | 35.20º N | 25.40º E | Rural/Remote | AMS | 25.09.2011 – 23.10.2011 | OOA (More-oxidized and Less-oxidized) |
| Mace Head | 53.33º N | 9.90º W | Rural/Remote | AMS | 01.01.2011 – 31.12.2011 | - |
| Marseille | 43.30º N | 5.40º E | Urban | AMS | 28.01.2011 – 02.03.2011 | HOA, BBOA, COA, OOA |
| Montsec | 42.03º N | 0.43º E | Rural/Remote | ACSM | 01.07.2011 – 01.12.2011 | HOA, LV-OOA, SV-OOA |
| SIRTA Paris | 48.71º N | 2.14º E | Suburban | ACSM | 01.10.2011 – 31.12.2011 | HOA, BBOA, COA-like, LV-OOA, SV-OOA |
| San Pietro Capofiume | 44.65º N | 11.62º E | Rural/Remote | AMS | 15.11.2011 – 01.12.2011 | HOA, BBOA, OOA |
| SMEAR II Hyytiälä | 61.85º N | 24.28º E | Rural/Remote | AMS | 14.03.2011 – 20.04.2011 | LV-OOA, SV-OOA |
| Zurich | 47.38º N | 8.53º E | Urban | ACSM | 11.02.2011 – 31.12.2011 | HOA, BBOA, COA, LV-OOA, SV-OOA |

[a] AMS: Aerodyne aerosol mass spectrometer; ACSM: Aerodyne aerosol chemical speciation monitor.

[b] LV-OOA: low-volatility oxygenated organic aerosol; SV-OOA: semi-volatile oxygenated organic aerosol.





**Table 2:** Statistical analysis of daily average organic aerosols calculated by the standard (BASE) and modified (NEW) parameterization at nine ACSM/AMS stations. MB: mean bias; ME: mean error; RMSE: root-mean-square error; MFB: mean fractional bias; MFE: mean fractional error.

| Stations | MB (µg m⁻³) | | MGE (µg m⁻³) | | RMSE (µg m⁻³) | | MFB(%) | | MFE(%) | |
|---|---|---|---|---|---|---|---|---|---|---|
| | BASE | NEW | BASE | NEW | BASE | NEW | BASE | NEW | BASE | NEW |
| OA | | | | | | | | | | |
| Bologna | -12.6 | -4.9 | 12.8 | 9.2 | 16.0 | 12.2 | -90 | -25 | 93 | 56 |
| Finokalia | -1.1 | -0.1 | 1.5 | 1.4 | 2.1 | 1.9 | -53 | -17 | 70 | 58 |
| Mace Head | -0.1 | 0.0 | 0.4 | 0.5 | 1.0 | 1.1 | -80 | -71 | 115 | 112 |
| Marseille | -5.8 | -3.5 | 5.8 | 4.0 | 7.8 | 6.2 | -102 | -46 | 105 | 60 |
| Montsec | -1.4 | -0.9 | 1.9 | 1.7 | 2.5 | 2.3 | -54 | -32 | 88 | 78 |
| Paris SIRTA | -6.2 | -3.4 | 6.3 | 4.5 | 9.5 | 7.3 | -109 | -39 | 113 | 66 |
| San Pietro Capofiume | -2.7 | 5.7 | 5.3 | 8.0 | 6.9 | 9.7 | -14 | 52 | 59 | 68 |
| SMEAR II | -0.1 | 0.6 | 0.5 | 0.8 | 0.9 | 1.3 | -15 | 53 | 55 | 69 |
| Zurich | -1.4 | 0.9 | 3.6 | 3.8 | 4.9 | 5.3 | -28 | 8 | 63 | 55 |
| HOA | | | | | | | | | | |
| Bologna | -2.2 | -0.6 | 2.2 | 1.8 | 3.5 | 2.8 | -88 | 7 | 100 | 70 |
| Finokalia [a] | - | - | - | - | - | - | - | - | - | - |
| Marseille | -0.8 | -0.1 | 0.9 | 0.9 | 1.4 | 1.4 | -79 | 6 | 115 | 85 |
| Montsec | -0.3 | -0.2 | 0.3 | 0.2 | 0.3 | 0.3 | -154 | -97 | 156 | 105 |
| Paris SIRTA | -0.6 | 0.1 | 0.6 | 0.6 | 1.2 | 1.1 | -74 | 24 | 99 | 76 |
| San Pietro Capofiume | -2.3 | -1.1 | 2.3 | 1.4 | 2.7 | 1.8 | -129 | -42 | 130 | 58 |
| SMEAR II [a] | - | - | - | - | - | - | - | - | - | - |
| Zurich | -0.2 | 0.5 | 0.4 | 0.7 | 0.6 | 1.0 | -29 | 62 | 81 | 85 |
| BBOA | | | | | | | | | | |
| Bologna | -4.2 | 0.2 | 4.3 | 3.0 | 5.4 | 3.7 | -95 | 7 | 97 | 50 |
| Finokalia | - | - | - | - | - | - | - | - | - | - |
| Marseille | -2.1 | -1.2 | 2.2 | 1.8 | 3.9 | 3.5 | -105 | -20 | 117 | 76 |
| Montsec | - | - | - | - | - | - | - | - | - | - |
| Paris SIRTA | -1.6 | 0.1 | 1.8 | 1.7 | 3.0 | 2.6 | -58 | 32 | 103 | 80 |
| San Pietro Capofiume | -1.0 | 4.1 | 2.9 | 5.3 | 4.3 | 6.5 | 35 | 104 | 115 | 118 |
| SMEAR II | - | - | - | - | - | - | - | - | - | - |
| Zurich | -0.5 | 0.2 | 0.6 | 0.8 | 0.9 | 1.2 | -74 | 15 | 98 | 74 |
| OOA | | | | | | | | | | |
| Bologna | -3.5 | -1.8 | 4.6 | 4.7 | 6.4 | 6.3 | -47 | -19 | 79 | 78 |
| Finokalia | -1.3 | -0.7 | 1.6 | 1.3 | 2.3 | 1.9 | -62 | -37 | 75 | 61 |
| Marseille | -2.5 | -1.8 | 2.6 | 2.1 | 3.3 | 2.7 | -77 | -48 | 87 | 65 |
| Montsec | -1.1 | -0.8 | 1.4 | 1.3 | 1.8 | 1.6 | -56 | -41 | 76 | 65 |
| Paris SIRTA | -2.6 | -2.2 | 2.6 | 2.3 | 4.1 | 3.8 | -111 | -91 | 117 | 102 |
| San Pietro Capofiume | 1.1 | 3.2 | 2.9 | 4.0 | 3.3 | 4.6 | 54 | 82 | 84 | 93 |
| SMEAR II | -0.3 | 0.0 | 0.5 | 0.5 | 0.9 | 0.9 | -53 | -19 | 80 | 66 |
| Zurich | 0.2 | 0.9 | 2.9 | 3.1 | 4.0 | 4.3 | -5 | 11 | 67 | 64 |

[a] HOA and BBOA were too low in Finokalia and SMEAR II to be resolved in the PMF analysis.





**Table 3:** Modelled relative contribution of different sources to total OA in eight sub-regions in Europe. DJF: December – January – February; JJA: June – July – August.

| OA sources | Iberian Peninsula | | The Mediterranean | | Po Valley | | Eastern Europe | | Central Europe | | Benelux | | Ireland and Great Britain | | Scandinavia | |
|---|---|---|---|---|---|---|---|---|---|---|---|---|---|---|---|---|
| | DJF | JJA | DJF | JJA | DJF | JJA | DJF | JJA | DJF | JJA | DJF | JJA | DJF | JJA | DJF | JJA |
| POA (%) | | | | | | | | | | | | | | | | |
| Gasoline vehicles | 0.2 | 0.1 | 0.6 | 0.5 | 1.8 | 1.3 | 0.2 | 0.2 | 0.3 | 0.2 | 0.4 | 0.3 | 0.3 | 0.2 | 0.2 | 0.1 |
| Diesel vehicles | 2.1 | 1.4 | 1.8 | 1.6 | 4.5 | 3.2 | 2.1 | 1.8 | 3.5 | 2.4 | 4.9 | 3.6 | 2.5 | 1.8 | 1.5 | 0.7 |
| Biomass burning | 26.5 | 7.6 | 35.2 | 7.2 | 52.6 | 10.6 | 52.4 | 13.4 | 38.4 | 6.7 | 37.9 | 7.4 | 16.3 | 3.0 | 42.8 | 3.8 |
| Other anthropogenic | 5.0 | 7.0 | 5.2 | 7.3 | 4.6 | 4.9 | 4.9 | 4.9 | 5.8 | 5.4 | 9.6 | 11.0 | 6.5 | 7.2 | 7.4 | 4.3 |
| **POA sum** | **33.8** | **16.2** | **42.8** | **16.6** | **63.5** | **20.0** | **59.5** | **20.3** | **48.0** | **14.8** | **52.8** | **22.4** | **25.6** | **12.1** | **51.8** | **8.9** |
| SOA (%) | | | | | | | | | | | | | | | | |
| Gasoline vehicles | 0.8 | 2.2 | 2.4 | 7.3 | 2.4 | 9.4 | 0.8 | 3.0 | 1.1 | 4.0 | 0.8 | 3.3 | 0.5 | 1.8 | 0.3 | 1.7 |
| Diesel vehicles | 0.3 | 0.8 | 0.5 | 1.5 | 0.4 | 1.7 | 0.3 | 1.2 | 0.4 | 1.4 | 0.3 | 1.2 | 0.1 | 0.5 | 0.1 | 0.5 |
| Biomass burning | 15.6 | 10.2 | 28.9 | 17.2 | 21.2 | 15.7 | 20.2 | 13.3 | 16.4 | 9.4 | 11.9 | 7.4 | 5.9 | 3.3 | 6.7 | 5.2 |
| Other anthropogenic | 4.7 | 15.5 | 6.4 | 22.2 | 4.4 | 21.3 | 2.7 | 11.2 | 3.8 | 16.1 | 3.0 | 16.5 | 1.9 | 11.8 | 1.1 | 6.8 |
| Biogenic | 44.7 | 55.2 | 19.0 | 35.2 | 8.1 | 31.9 | 16.5 | 51.0 | 30.2 | 54.3 | 31.1 | 49.2 | 66.0 | 70.4 | 39.9 | 77.0 |
| **SOA sum** | **66.2** | **83.8** | **57.2** | **83.4** | **36.5** | **80.0** | **40.5** | **79.7** | **52.0** | **85.2** | **47.2** | **77.6** | **74.4** | **87.9** | **48.2** | **91.1** |



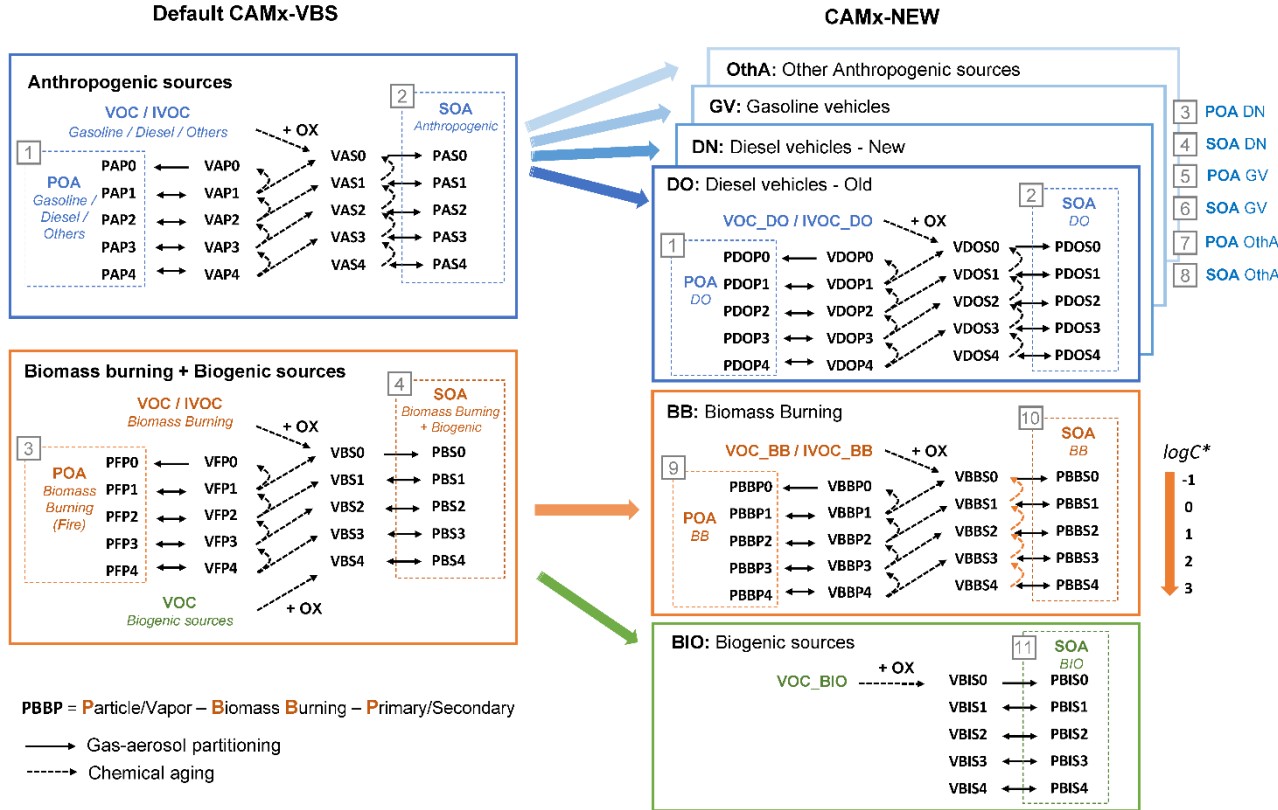

**Figure 1**: Schematic diagram of the default (left, with 4 sets) and modified VBS module with extended basis sets (right, with 11 sets). The number 0 – 4 in the species names present the five volatility bins ranging from $10^{-1}$ to $10^3$ µg m$^{-3}$ in saturation vapour concentration (C*) at 298 K. The numbers in the grey boxes indicate the number of basis sets (4 sets for default CAMx–VBS, and 11 sets for CAMx–NEW). The primary organic aerosol (POA) components are oxidized to POA and SOA in the next lower volatility bins (shown as dashed arrows). The same reaction scheme but different volatility distribution and yields parameters are adopted for the anthropogenic sources DO, DN, GV and OthA. The parameterization of DO, GV and OthA follows the default values in CAMx–VBS. The modified parameterization CAMx–NEW enables the aging of secondary biomass burning vapour (brown dashed arrows in the BB box). The set of cooking (PCP) is not presented in CAMx–VBS due to missing cooking emission input in this study.





**Figure 2**: Temporal variation of modelled (with both BASE and NEW parameterization) and measured organic aerosol concentration together with some meteorological parameters available at each station (dotted line: measurements, solid line: model). Hourly OA concentrations was used for San Pietro Capofiume, while daily average OA concentration was used for other stations.



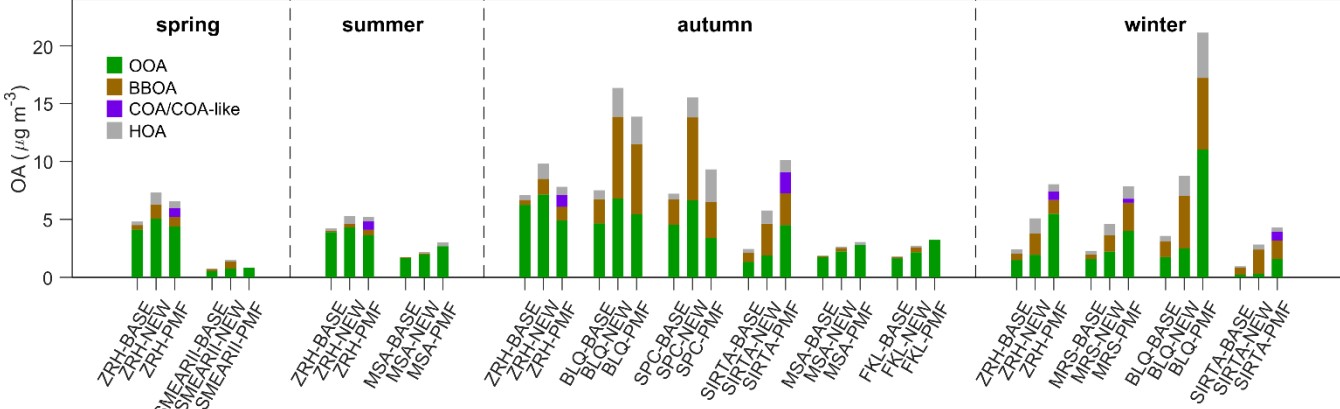

**Figure 3:** Seasonal variation of PMF analysis results and modelled OA components (with both BASE and NEW
parameterization). The average of modelled OA components was calculated based on the data during the same periods with
measurements at each stations as shown in Table 1. ZRH: Zurich, SMEARII: SMEAR II Hyytiälä, MSA: Montsec, BLQ:
Bologna, SPC: San Pietro Capofiume, SIRTA: Paris SIRTA, FKL: Finokalia, MRS: Marseille. Spring: March–April–May,
summer: June–July–August, autumn: September–October–November, winter: December–January (2011)–February (2011).





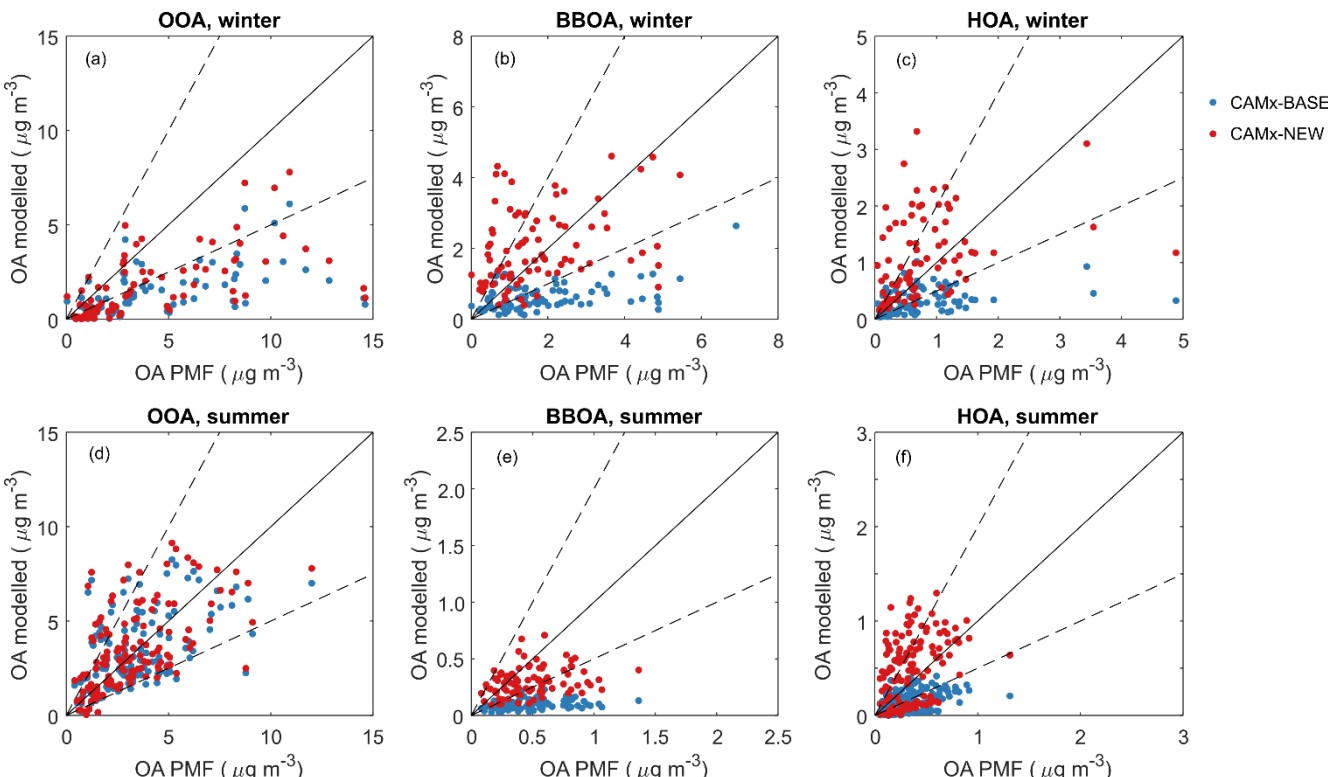

**Figure 4:** Comparison between modelled daily average and PMF analysis results in winter and summer. The summer data include measurements at Zurich and Montsec, and the winter data include measurements at Zurich, Marseille, Bologna and SIRTA.







**Figure 5:** Comparison between modelled relative contributions of OA components and positive matrix factorization (PMF) analysis results at Zurich (urban site) and Montsec (remote site). GV: Gasoline Vehicles; DV: Diesel Vehicles; BB: Biomass Burning; OP: Other anthropogenic sources; BIO: Biogenic sources.





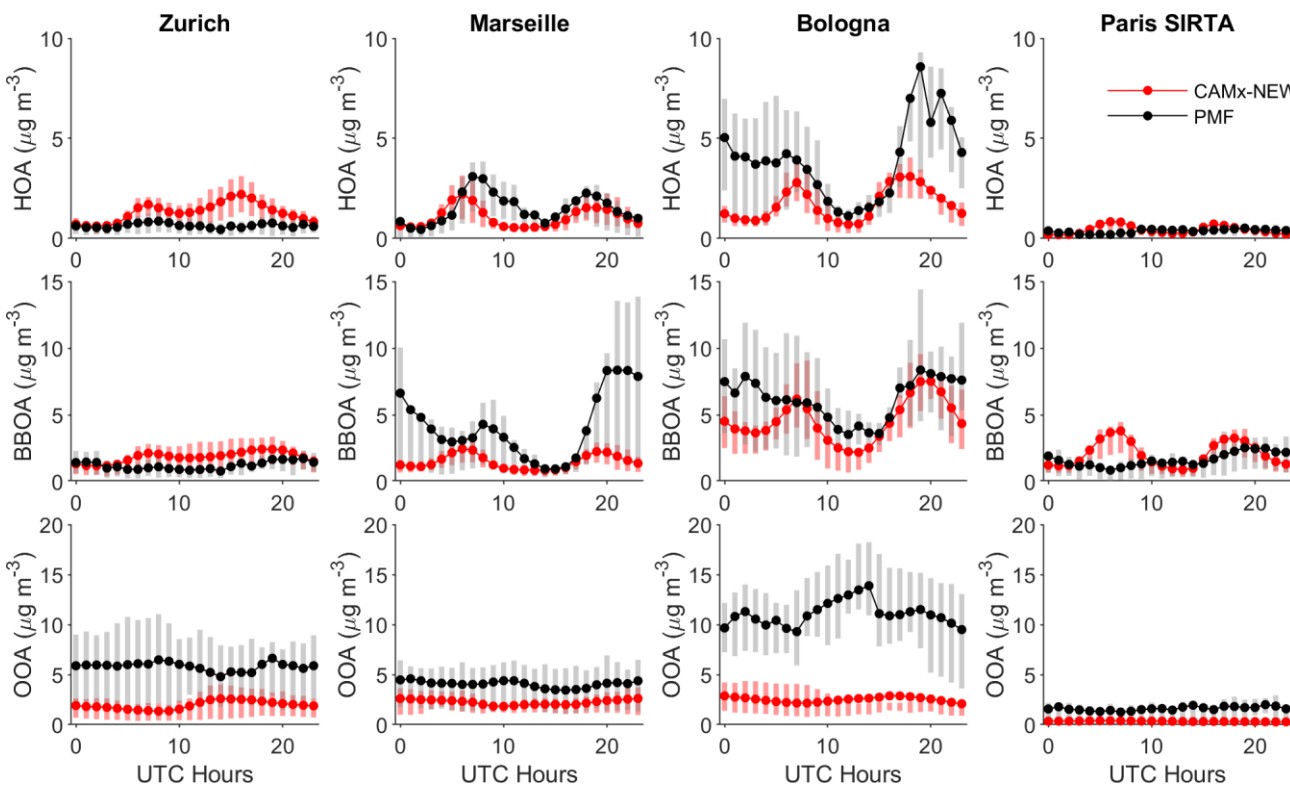

**Figure 6:** Diurnal variations of modelled OA components by CAMx-NEW and PMF studies at urban sites in winter. The lines represent the average value of modelled and measured OA components during the same periods for each stations (as displayed in Figure 2b), and the bars represent 25th and 75th percentiles of hourly data.





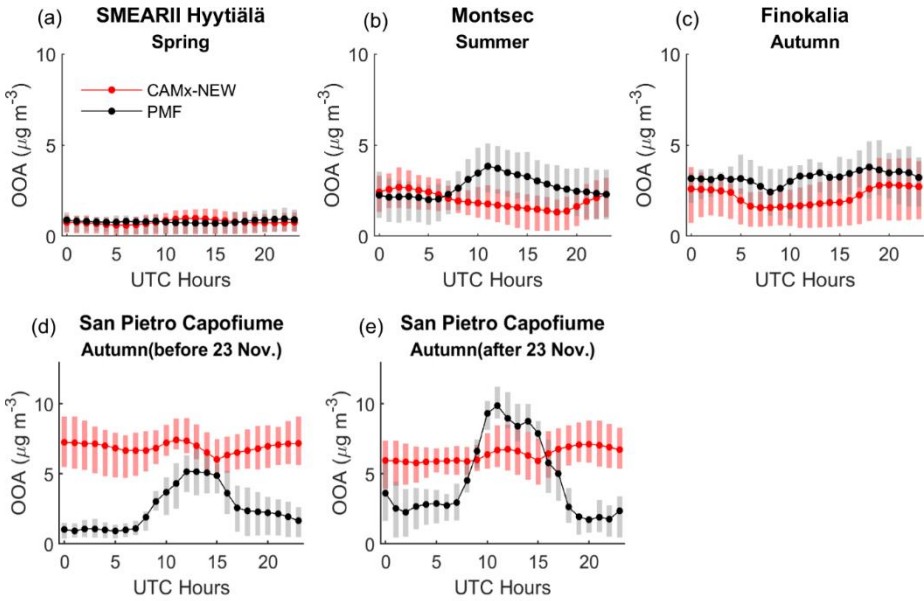

**Figure 7:** Diurnal variations of modelled OOA components by CAMx-NEW and PMF studies in rural or remote sites. The
lines represent the average value of modelled and measured OOA. Bars represent 25th and 75th percentiles of hourly data.





**Figure 8:** Relative contributions of different sources (POA and SOA) to total OA in winter (a, b) and summer (c, d). The winter and summer results are the averages of December – January – February and June – July – August, respectively. OA from biogenic sources is all secondary so the POA panels for biogenic sources are empty.



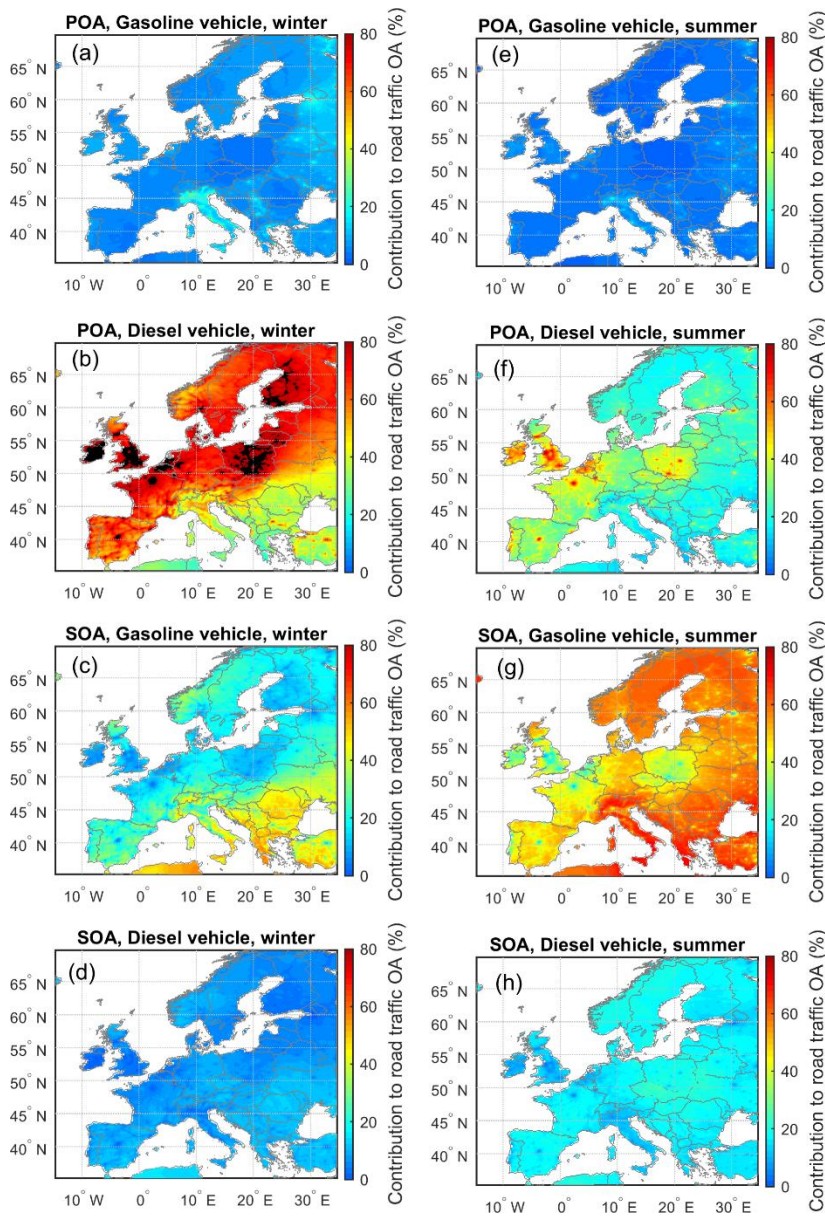

**Figure 9:** Contributions of gasoline and diesel vehicles to the road traffic (gasoline + diesel vehicles) OA concentration in Europe during winter (December – January – February) and summer (June – July – August). .

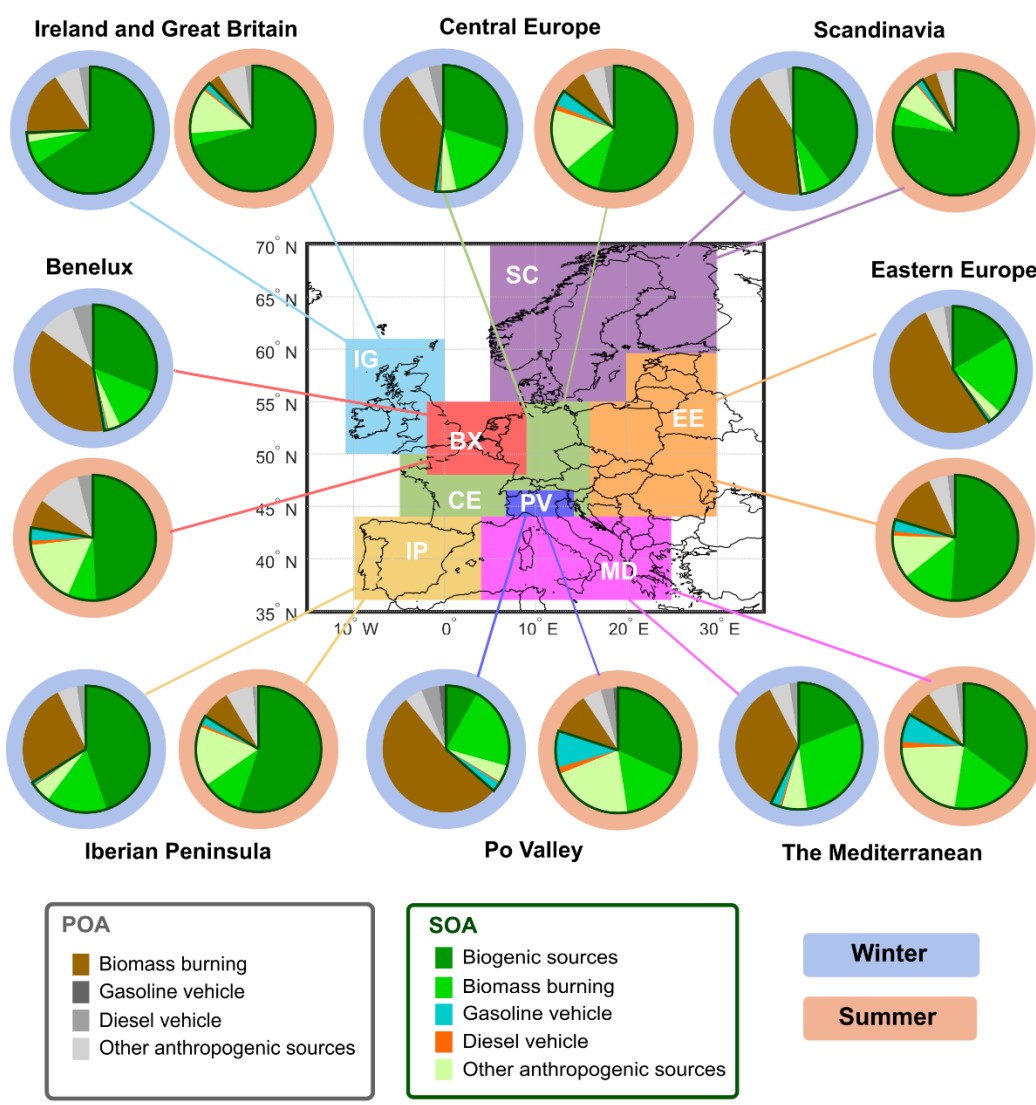

**Figure 10:** Modelled regional variations of primary and secondary organic aerosols sources in Europe in winter (December – January – February) and summer (June – July – August). The 8 sub-regions are the Iberian Peninsula (IP), the Mediterranean (MD), Po Valley (PV), eastern Europe (EE), central Europe (CE), Benelux (BX), Ireland and Great Britain (IG) and Scandinavia (SC).

