# Peer review of "Sources of organic aerosols in Europe: A modelling study using CAMx with modified volatility basis set scheme"

_Atmospheric Chemistry and Physics, 2019_

## Referee Comment (RC1) · Anonymous Referee #2 · 10 Jul 2019

General comments The paper "Sources of organic aerosols in Europe: A modelling study using CAMx with modified volatility basis set scheme" by Jiang et al. deals with a very interesting topic for modelling science. Indeed, the modelling reconstruction of Organic aerosol fraction remains a challenging issue due to the relevant number of species and processes involved. The paper can surely provide a very interesting contribution to the scientific knowledge in this field, particularly in the European context and therefore fits the scope of ACP. The paper is well written, with concise and clear statements, and it does not require any substantial review of syntax and language.

However, before publication, there are a few issues that should be addressed by the

authors, that are detailed in the following:

1) One of the key aspect of the paper concerns the implementation of modified parameterizations in the 1.5 VBS scheme, however from section 2.2.3 is not clear what modifications have been actually introduced beyond the split of the original 5 basis sets into 11. From the text it seems that just two modifications were introduced: a. Setting SOA yield for DN to 0. b. Enabling oxidation of SOA from Biomass burning Is it correct? If yes, the modifications introduced by authors are surely reasonable and interesting, but limited only to a few aspects of 1.5 VBS scheme and this should be better clarified in the text. Conversely if other modifications have been introduced they should be better describe it (maybe introducing a table comparing BASE and NEW VBS parameterizations)

2) Authors point out that one of added values of their work is the evaluation of model results over a long term period. However both meteorological and air quality model performance evaluation (for chemical species other than OA) is limited only to a winter and summer month. Though interesting, such analysis is not fully adequate to evaluate the CAMx performance over the whole year. Moreover, in most cases the selected months (February and July) do not overlap with the observation periods of OA measurements (see table 1). An yearly based analysis of AQ and meteorological model performance should be added. Moreover, considering the observation periods covered by OA measurements a seasonal based analysis could be added too. The latter would also be coherent with several results presented by authors in sections 3.1.2, 3.2, 3.3 and 3.4

3) The "NEW" simulation includes two main modifications, the first one concerning the VBS scheme and the second one related to the estimation of SVOC emissions. It would be very interesting introducing an intermediate simulation where only one the two modifications is implemented (e.g. only the modified VBS scheme either only the chance in SVOC emissions). This could help in better quantifying the contribution of every change to the total concentration variation. The results of such analysis could be

introduced in Table 2 as well as in figure 2 and 3.

4) The analysis of the obtained results allows the authors to conclude that the "NEW" introduces an overall improvement of CAMx performance. I fully agree with their conclusion, however SOA performance still highlights a general underestimation , differently from HOA and BBOA that are reproduced fairly well in "NEW" run. Could this result be specifically related to a possible underestimation of some key precursors such as IVOC? Any comment/additional analysis of this issue?

Specific comments P7 R15-16 Were coarse PM emissions split into EC, Na and SO4=? The default CAMx aerosol scheme (CF) includes only CCRS and CPRM species for the coarse fraction. In case, what aerosol scheme was used?

P8 R5-8 SVOC emissions play a key role on OA processes both in terms of total mass as well as with respect to their volatility distribution. Did the authors introduce also a different volatility distribution in the "NEW" run, beyond increasing the total emissions by a factor of 3?

P8 R22-24 Authors correctly point out that NO2 is underestimated suggesting that the observed discrepancy could be related to a corresponding underestimation of NOx emissions. As NOx emissions are mostly related to road transport could other emissions of the same sector be underestimated too? (e.g. NMVOC...). Any possible influence on OM results?

P9 R24-30 CAMx performance in reproducing SO2 are rather poor. Modelled concentrations are strongly overestimated (MFB is higher than 75% in February) and substantially uncorrelated to observed values (IOA is around 0.1). Considering that SO2 emissions are mostly related to "Other anthropogenic sources" do authors think that such overestimation could influence also other species (e.g. NMVOC, PM) and, therefore, also the contribution of this sector to OM concentrations? (see for example conclusions P16 R5-9)

[Figure]

P10 R1-2 Table 2 is a bit misleading because it summarizes the model performance at all ACSM/AMS stations, which cover very different temporal periods. The different rows should be grouped somehow, for example separating stations covering the whole year from sites covering only winter periods, summer periods, etc.. Moreover, it could be useful adding, for each site, the number of available observations, as well as the observation period (though already reported in table I)

Technical corrections P19 R18 Table 1 ?

---

## Referee Comment (RC2) · Manish Shrivastava (Referee) · 5 Oct 2019

Jiang et al. provide a modeling study of organic aerosols in Europe using a volatility basis set approach. This study is important for this region and provides good insights about sources and formation of OA. Results are evaluated with measurements including PMF analysis of AMS/ACSM data.

Below I have several suggestions for improvement and also citation of relevant papers that need to be considered by the authors.

General comments on modified VBS approach: VBS is a framework that represents

gas-particle partitioning and multigenerational aging of SOA. But depending on SIVOC emissions, reaction rates, functionalization/fragmentation branching etc. different implementations of VBS can produce very different results. Thus, it is important to describe VBS developments in the context of previous studies, specifically acknowledging and documenting differences. The authors describe their VBS as a modified VBS approach. But use of a "modified VBS" terminology has been used in 2 previous papers from M. Shrivastava et al. 2013, 2015. Those papers included both functionalization and fragmentation of organics and compared model results to several field measurements (surface based and aircraft measurements). See:

https://agupubs.onlinelibrary.wiley.com/doi/full/10.1002/jgrd.50160
https://agupubs.onlinelibrary.wiley.com/doi/full/10.1002/2014JD022563

The authors use the 1.5D VBS from Koo et al. To avoid confusion between author's version of VBS and previous 2 papers (above), I recommend the authors add a few sentences about how their modified VBS differs from the modified VBS aging parameterizations developed by M. Shrivastava et al. It may be better to refer to their VBS as 1.5D VBS, since this is what they used. It would be also instructive to compare their modified VBS results with those from M. Shrivastava et al. Note that Cholakian et al. 2018 (cited in this paper) used a similar modified VBS as Shrivastava et al. 2013,2015.

Specific comments: Page 3 Line 5-10: In addition to Hallquist et al. 2009, also cite M. Shrivastava et al. 2017 Review paper on SOA published in Reviews of Geophysics: https://agupubs.onlinelibrary.wiley.com/doi/full/10.1002/2016RG000540

Page 3 Line 15-20: For WRF-Chem please cite 2 of the more recent papers on VBS implementation of SOA in addition to Shrivastava et al. 2011: https://www.nature.com/articles/s41467-019-08909-4 https://agupubs.onlinelibrary.wiley.com/doi/full/10.1002/jgrd.50160

Page 7 Line 5: While several models underpredict OA from biomass burning, some models predict OA from biomass burning could be much more important . See

https://agupubs.onlinelibrary.wiley.com/doi/full/10.1002/2014JD022563.   This should be acknowledged here in addition to Hodzic et al. 2010 for biomass burning.

Page 11: Aqueous chemistry of organic aerosols (OA) in fog can also increase OA by 4-20% (see Gilardoni et al. 2014 PNAS for Po Valley Italy measurements of aqueous SOA). Since the authors underestimate winter-time OOA, missing aqueous phase SOA in fog would be an important source. Although they are overestimating OA during the autumn due to modeled bias in relative humidity and wet scavenging, the high bias could be due to other reasons like overestimation of SIVOC emissions, biases in aging paramaterizations. This needs to be acknowledged as a caveat. The authors could also compare rain rates simulated by their model to measurements in that region to provide further evidence for model underestimation of rain rate/wet scavenging.

Page 12 Above line 5: Instead of "biomass density" the authors could probably just say increasing biogenic emissions here?

Page 12 Line 20: From Figure 6 it seems the authors could have applied a site specific scaling of POA emissions based on PMF HOA+BBOA. This could improve their POA, its diurnal variation and also IVOC emissions for biomass burning that are calculated as ~4 times BB-POA. Please comment on use of a site-specific scaling of BB-POA and IVOC emissions based on PMF HOA+BBOA.

---

## Author Comment (AC1) · 28 Oct 2019

**Responses to the comments of anonymous referee #1**

We thank the referee for the valuable comments that have greatly helped us to improve the manuscript. Please find below our responses (in black) after the referee comments (in blue). The changes in the revised manuscript are written in *italic*.

**General comments**

The paper "Sources of organic aerosols in Europe: A modelling study using CAMx with modified volatility basis set scheme" by Jiang et al. deals with a very interesting topic for modelling science. Indeed, the modelling reconstruction of Organic aerosol fraction remains a challenging issue due to the relevant number of species and processes involved. The paper can surely provide a very interesting contribution to the scientific knowledge in this field, particularly in the European context and therefore fits the scope of ACP. The paper is well written, with concise and clear statements, and it does not require any substantial review of syntax and language.

However, before publication, there are a few issues that should be addressed by the authors, that are detailed in the following:

1) One of the key aspect of the paper concerns the implementation of modified parameterizations in the 1.5 VBS scheme, however from section 2.2.3 is not clear what modifications have been actually introduced beyond the split of the original 5 basis sets into 11. From the text it seems that just two modifications were introduced: a) Setting SOA yield for DN to 0. b) Enabling oxidation of SOA from Biomass burning. Is it correct? If yes, the modifications introduced by authors are surely reasonable and interesting, but limited only to a few aspects of 1.5 VBS scheme and this should be better clarified in the text. Conversely if other modifications have been introduced they should be better describe it (maybe introducing a table comparing BASE and NEW VBS parameterizations)

The major modification in the code was extending and splitting the standard VBS scheme to enable source apportionment for POA and SOA, together with the adjustment of yields for the new diesel vehicles and oxidation rate of SOA from biomass burning. Since most of the changes are in the model framework (adding new species to the model species list, splitting reactions for precursors from different sources), it is difficult to display them in a table. We will however upload the modified model codes in a public data repository once published. Following similar comments of referee 2#, we added a more detailed description of the 1.5-D VBS and our modifications in this study which we called PSI-VBS in order to distinguish from other modified versions.

[revised manuscript text omitted]

2) Authors point out that one of added values of their work is the evaluation of model results over a long term period. However both meteorological and air quality model performance evaluation (for chemical species other than OA) is limited only to a winter and summer month. Though interesting, such analysis is not fully adequate to evaluate the CAMx performance over the whole year. Moreover, in most cases the selected months (February and July) do not overlap with the observation periods of OA measurements (see table 1). An yearly based analysis of AQ and meteorological model performance should be added. Moreover, considering the observation periods covered by OA measurements a seasonal based analysis could be added too. The latter would also be coherent with several results presented by authors in sections 3.1.2, 3.2, 3.3 and 3.4.

A whole-year model performance evaluation for both meteorological and air quality simulations was added into the revised manuscript. Statistical results for different seasons (December-January-February, March-April-May, June-July-August, and September-October-November) as well as for whole year were displayed in Tables S2 and S3. The numbers in section 3.1.1 were updated as well.

As the referee mentioned, actually we have already presented a season-based analysis for OA components in section 3.2. To be more coherent, we added the Table S5 presenting the seasonal statistical results as a supplement to Table 2 and Figure 3.

***Table S5.*** *Seasonal statistical analysis of daily average organic aerosols at nine ACSM/AMS stations. MB: mean bias; ME: mean error; RMSE: root-mean-square error; MFB: mean fractional bias; MFE: mean fractional error. Spring: March-April-May, summer: June-July-August, autumn: September-October-November, winter: December-January-February.*

| Season | Site | MB ($\mu g\ m^{-3}$) | | ME ($\mu g\ m^{-3}$) | | RMSE ($\mu g\ m^{-3}$) | | MFB (%) | | MFE (%) | |
|---|---|---|---|---|---|---|---|---|---|---|---|
| | | BASE | NEW | BASE | NEW | BASE | NEW | BASE | NEW | BASE | NEW |
| **OA** | | | | | | | | | | | |
| spring | MHD | -0.1 | 0.1 | 0.7 | 0.8 | 1.6 | 1.7 | -86 | -74 | 117 | 114 |
| | MRS | -9.4 | -7.6 | 9.4 | 7.6 | 9.7 | 8.0 | -163 | -116 | 163 | 116 |
| | ZRH | -1.7 | 0.8 | 3.5 | 3.7 | 4.7 | 5.0 | -34 | 8 | 64 | 53 |
| | SMEARII | 0.0 | 0.7 | 0.4 | 0.8 | 0.6 | 1.2 | -10 | 58 | 54 | 70 |
| summer | MHD | 0.0 | 0.0 | 0.3 | 0.3 | 0.5 | 0.6 | -91 | -83 | 119 | 115 |
| | MSA | -2.4 | -2.0 | 2.6 | 2.3 | 3.2 | 2.8 | -89 | -73 | 97 | 83 |
| | ZRH | -1.0 | 0.1 | 3.2 | 3.3 | 4.8 | 5.0 | -21 | 3 | 65 | 59 |
| autumn | BLQ | -8.4 | 0.5 | 8.7 | 5.5 | 10.8 | 7.0 | -62 | 10 | 67 | 35 |
| | FKL | -1.1 | -0.1 | 1.5 | 1.4 | 2.2 | 1.9 | -53 | -17 | 70 | 58 |
| | MHD | -0.1 | -0.1 | 0.2 | 0.2 | 0.5 | 0.5 | -90 | -83 | 116 | 113 |
| | MSA | -1.4 | -0.6 | 1.9 | 1.7 | 2.4 | 2.2 | -42 | -12 | 72 | 59 |
| | SIRTA | -7.6 | -4.3 | 7.7 | 5.4 | 11.0 | 8.5 | -108 | -43 | 111 | 63 |
| | SPC | -2.7 | 5.7 | 5.3 | 8.0 | 6.9 | 9.6 | -14 | 52 | 59 | 68 |
| | ZRH | -0.6 | 2.2 | 3.6 | 4.4 | 4.7 | 6.1 | -16 | 17 | 54 | 53 |
| winter | BLQ | -21.4 | -16.2 | 21.4 | 16.4 | 23.4 | 19.0 | -149 | -98 | 149 | 99 |
| | MHD | 0.0 | 0.1 | 0.3 | 0.3 | 0.6 | 0.6 | -4 | 7 | 91 | 89 |
| | MRS | -7.8 | -5.3 | 7.8 | 5.8 | 11.0 | 9.3 | -109 | -55 | 111 | 65 |
| | MSA | 0.1 | 0.5 | 0.6 | 0.9 | 0.9 | 1.3 | -10 | 10 | 90 | 92 |
| | SIRTA | -3.6 | -1.7 | 3.6 | 2.8 | 5.5 | 4.4 | -110 | -31 | 117 | 71 |
| | ZRH | -4.1 | -0.5 | 4.6 | 3.7 | 5.9 | 4.9 | -68 | -3 | 82 | 55 |
| **HOA** | | | | | | | | | | | |
| spring | MRS | -1.4 | -0.8 | 1.4 | 1.2 | 2.1 | 1.7 | -107 | -33 | 137 | 99 |
| | ZRH | -0.3 | 0.4 | 0.4 | 0.7 | 0.6 | 0.9 | -41 | 52 | 81 | 81 |
| summer | MSA | -0.3 | -0.2 | 0.3 | 0.2 | 0.3 | 0.3 | -157 | -102 | 158 | 111 |
| | ZRH | -0.2 | 0.3 | 0.3 | 0.5 | 0.4 | 0.6 | -29 | 61 | 80 | 83 |
| autumn | BLQ | -1.6 | 0.2 | 1.7 | 1.4 | 2.6 | 1.9 | -66 | 35 | 82 | 65 |
| | MSA | -0.2 | -0.2 | 0.2 | 0.2 | 0.3 | 0.2 | -147 | -84 | 149 | 92 |
| | SIRTA | -0.7 | 0.1 | 0.8 | 0.8 | 1.4 | 1.3 | -76 | 21 | 98 | 73 |
| | SPC | -2.3 | -1.1 | 2.3 | 1.4 | 2.7 | 1.8 | -129 | -42 | 130 | 58 |
| | ZRH | -0.3 | 0.6 | 0.5 | 0.9 | 0.7 | 1.1 | -26 | 65 | 81 | 86 |
| winter | BLQ | -3.4 | -2.2 | 3.4 | 2.6 | 4.9 | 4.1 | -134 | -53 | 137 | 82 |
| | MRS | -1.2 | -0.5 | 1.3 | 1.1 | 2.1 | 1.8 | -88 | -3 | 116 | 83 |
| | SIRTA | -0.3 | 0.1 | 0.3 | 0.3 | 0.5 | 0.5 | -70 | 30 | 101 | 80 |
| | ZRH | -0.1 | 1.0 | 0.4 | 1.2 | 0.6 | 1.5 | 3 | 92 | 80 | 105 |
| **BBOA** | | | | | | | | | | | |
| spring | MRS | -3.8 | -3.0 | 3.8 | 3.0 | 4.6 | 3.9 | -164 | -104 | 164 | 104 |
| | ZRH | -0.4 | 0.4 | 0.6 | 0.8 | 0.8 | 1.2 | -57 | 37 | 85 | 74 |
| summer | ZRH | -0.4 | -0.2 | 0.4 | 0.3 | 0.6 | 0.5 | -108 | -28 | 120 | 75 |
| autumn | BLQ | -4.0 | 1.0 | 4.0 | 2.9 | 5.1 | 3.5 | -83 | 23 | 86 | 46 |
| | SIRTA | -2.0 | 0.0 | 2.1 | 1.9 | 3.4 | 2.8 | -70 | 22 | 103 | 76 |
| | SPC | -1.0 | 4.1 | 2.9 | 5.3 | 4.3 | 6.5 | 35 | 104 | 115 | 118 |
| | ZRH | -0.8 | 0.2 | 0.8 | 0.8 | 1.2 | 1.2 | -74 | 18 | 94 | 67 |
| winter | BLQ | -4.8 | -1.6 | 4.8 | 3.2 | 6.0 | 4.2 | -121 | -29 | 121 | 59 |
| | MRS | -3.6 | -2.6 | 3.7 | 3.1 | 6.9 | 6.4 | -118 | -37 | 126 | 80 |
| | SIRTA | -1.0 | 0.5 | 1.2 | 1.5 | 2.1 | 2.1 | -46 | 46 | 100 | 85 |
| | ZRH | -0.5 | 1.3 | 0.8 | 1.7 | 1.2 | 2.2 | -21 | 71 | 82 | 96 |
| **OOA** | | | | | | | | | | | |
| spring | MRS | -3.9 | -3.5 | 3.9 | 3.5 | 3.9 | 3.6 | -158 | -135 | 158 | 135 |
| | ZRH | -0.3 | 0.6 | 2.9 | 3.0 | 3.8 | 4.0 | -15 | 6 | 69 | 64 |
| | SMEARII | -0.2 | 0.0 | 0.4 | 0.4 | 0.6 | 0.6 | -57 | -23 | 81 | 65 |
| summer | MSA | -1.1 | -0.9 | 1.5 | 1.3 | 1.9 | 1.7 | -62 | -48 | 82 | 72 |
| | ZRH | 0.4 | 0.8 | 2.5 | 2.5 | 3.5 | 3.6 | 5 | 16 | 64 | 62 |

| Season | Site | MB (µg m$^{-3}$) | | ME (µg m$^{-3}$) | | RMSE (µg m$^{-3}$) | | MFB (%) | | MFE (%) | |
|---|---|---|---|---|---|---|---|---|---|---|---|
| | | BASE | NEW | BASE | NEW | BASE | NEW | BASE | NEW | BASE | NEW |
| autumn | BLQ | -0.8 | 1.4 | 2.3 | 2.9 | 2.9 | 3.5 | -2 | 33 | 49 | 54 |
| | FKL | -1.3 | -0.7 | 1.6 | 1.3 | 2.3 | 1.9 | -62 | -37 | 75 | 61 |
| | MSA | -1.0 | -0.5 | 1.3 | 1.1 | 1.7 | 1.5 | -43 | -23 | 61 | 49 |
| | SIRTA | -3.2 | -2.6 | 3.2 | 2.9 | 4.8 | 4.4 | -97 | -72 | 104 | 86 |
| | SPC | 1.1 | 3.2 | 2.9 | 4.0 | 3.3 | 4.6 | 54 | 82 | 84 | 93 |
| | ZRH | 1.5 | 2.4 | 3.0 | 3.4 | 4.2 | 4.9 | 17 | 31 | 58 | 60 |
| winter | BLQ | -9.3 | -8.5 | 9.3 | 8.5 | 10.4 | 9.8 | -144 | -126 | 144 | 126 |
| | MRS | -2.5 | -1.8 | 2.6 | 2.2 | 3.4 | 2.9 | -71 | -43 | 84 | 65 |
| | SIRTA | -1.3 | -1.3 | 1.4 | 1.3 | 2.0 | 2.0 | -138 | -127 | 142 | 133 |
| | ZRH | -3.8 | -3.1 | 4.2 | 3.8 | 5.8 | 5.4 | -78 | -55 | 98 | 86 |

3) The "NEW" simulation includes two main modifications, the first one concerning the VBS scheme and the second one related to the estimation of SVOC emissions. It would be very interesting introducing an intermediate simulation where only one the two modifications is implemented (e.g. only the modified VBS scheme either only the chance in SVOC emissions). This could help in better quantifying the contribution of every change to the total concentration variation. The results of such analysis could be introduced in Table 2 as well as in figure 2 and 3.

The intermediate simulation only changing the SVOC emissions (referred to as 3POA) can be seen in the Figure 1 below. The modelled land-average SOA from biomass burning in winter increased by 76% and 28% using NEW and 3POA compared to BASE, respectively. However, the influence of each modification varies depending on both location and time period. For the measurement sites in this study, NEW generally leads to a better model performance than 3POA, but the difference between 3POA and NEW is small (see Table 2 below). However, the difference is higher at locations where biomass burning is the dominant source. That is why we decided to present only BASE and NEW in Table 2 and Figure 2, 3 to avoid any misunderstanding such as further aging of biomass burning SOA is not important.

[Figure]

**Figure 1** Comparison of modelled winter (December-January-February) SOA from biomass burning by BASE (a), 3POA (b) and NEW (c) parameterization. The $con_{avg}$ values indicate the average concentration for the land.

**Table 2** Statistical results for modelled winter OA and OOA by different OA scheme.

| Site | MB (µg m$^{-3}$) | | | ME (µg m$^{-3}$) | | | RMSE (µg m$^{-3}$) | | | MFB (%) | | | MFE (%) | | |
|---|---|---|---|---|---|---|---|---|---|---|---|---|---|---|---|
| | BASE | 3POA | NEW | BASE | 3POA | NEW | BASE | 3POA | NEW | BASE | 3POA | NEW | BASE | 3POA | NEW |
| **OA** | | | | | | | | | | | | | | | |
| BLQ | -21.4 | -16.6 | -16.2 | 21.4 | 16.8 | 16.4 | 23.4 | 19.3 | 19.0 | -149 | -101 | -98 | 149 | 102 | 99 |
| MHD | 0.0 | 0.1 | 0.1 | 0.3 | 0.3 | 0.3 | 0.6 | 0.6 | 0.6 | -4 | 7 | 7 | 91 | 89 | 89 |
| MRS | -7.8 | -5.8 | -5.3 | 7.8 | 6.1 | 5.8 | 11.0 | 9.6 | 9.3 | -109 | -64 | -55 | 111 | 72 | 65 |
| MSA | 0.1 | 0.4 | 0.5 | 0.6 | 0.8 | 0.9 | 0.9 | 1.1 | 1.3 | -10 | 7 | 10 | 90 | 90 | 92 |
| SIRTA | -3.6 | -1.7 | -1.7 | 3.6 | 2.8 | 2.8 | 5.5 | 4.4 | 4.4 | -110 | -31 | -31 | 117 | 71 | 71 |
| ZRH | -4.1 | -0.9 | -0.5 | 4.6 | 3.7 | 3.7 | 5.9 | 4.9 | 4.9 | -68 | -8 | -3 | 82 | 56 | 55 |
| **OOA** | | | | | | | | | | | | | | | |
| BLQ | -9.3 | -8.9 | -8.5 | 9.3 | 8.9 | 8.5 | 10.4 | 10.1 | 9.8 | -144 | -133 | -126 | 144 | 133 | 126 |
| MRS | -2.5 | -2.2 | -1.8 | 2.6 | 2.4 | 2.2 | 3.4 | 3.2 | 2.9 | -71 | -60 | -43 | 84 | 76 | 65 |
| SIRTA | -1.3 | -1.3 | -1.3 | 1.4 | 1.3 | 1.3 | 2.0 | 2.0 | 2.0 | -138 | -129 | -127 | 142 | 134 | 133 |
| ZRH | -3.8 | -3.4 | -3.1 | 4.2 | 4.0 | 3.8 | 5.8 | 5.6 | 5.4 | -78 | -64 | -55 | 98 | 91 | 86 |

4) The analysis of the obtained results allows the authors to conclude that the "NEW" introduces an overall improvement of CAMx performance. I fully agree with their conclusion, however SOA

performance still highlights a general underestimation, differently from HOA and BBOA that are reproduced fairly well in "NEW" run. Could this result be specifically related to a possible underestimation of some key precursors such as IVOC? Any comment/additional analysis of this issue?
Although we tried to improve the model performance for SOA by adding the SVOC emissions and aging of biomass burning SOA, there are still a number of possible reasons for the underestimation: It could be the underestimated IVOC/SVOC emissions as the referee mentioned, or missing pathways of SOA formation such as aqueous-phase chemistry or uncertainties of the VBS parameters (yield, reaction rate, *etc.*) due to wall loss, etc. We added a paragraph in Section 3.2 (P13 L18 – L31) as a summary of the possible reasons and future work.

[revised manuscript text omitted]

P8 R5-8 SVOC emissions play a key role on OA processes both in terms of total mass as well as with respect to their volatility distribution. Did the authors introduce also a different volatility distribution in the "NEW" run, beyond increasing the total emissions by a factor of 3?

We totally agree with the referee that the volatility distribution is important for the issue of SVOC. For this study, we mainly focused on the total mass, and therefore the default volatility distribution was used for "NEW" run. However, we have already constrained the volatility distribution for SVOCs based on recent experimental studies (Stefenelli et al., 2019) and it will be implemented in CAMx soon as the next step.

Stefenelli, G., Jiang, J., Bertrand, A., Bruns, E. A., Pieber, S. M., Baltensperger, U., Marchand, N., Aksoyoglu, S., Prévôt, A. S. H., Slowik, J. G., and El Haddad, I.: Secondary organic aerosol formation from smoldering and flaming combustion of biomass: a box model parametrization based on volatility basis set, Atmos. Chem. Phys., 19, 11461–11484, https://doi.org/10.5194/acp-19-11461-2019, 2019.

P9 R22-24 Authors correctly point out that $NO_2$ is underestimated suggesting that the observed discrepancy could be related to a corresponding underestimation of $NO_x$ emissions. As $NO_x$ emissions are mostly related to road transport, could other emissions of the same sector be underestimated too? (e.g. NMVOC). Any possible influence on OM results?

One cannot generally assume that different components from the same sector are likely underestimated when this is shown for one of the components. Specifically for NMVOC and $NO_x$ from traffic, the emission sources are due to different types of vehicles and different driving conditions. $NO_x$ emissions are dominated by diesel vehicles during high load and high accelerations, while most NMVOC are due to gasoline emissions during cold start (Platt et al., 2017). At most sites, the contribution of traffic SOA is rather minor. However, we cannot rule out that especially under cold temperatures VOC emissions during cold start are especially high. Available [14]C measurements do not indicate large missing fossil organic carbon sources.

Platt, S. M., El Haddad, I., Pieber, S. M., Zardini, A. A., Suarez-Bertoa, R., Clairotte, M., Daellenbach, K. R., Huang, R. J., Slowik, J. G., Hellebust, S., Temime-Roussel, B., Marchand, N., de Gouw, J., Jimenez, J. L., Hayes, P. L., Robinson, A. L., Baltensperger, U., Astorga, C., and Prévôt, A. S. H.: Gasoline cars produce more carbonaceous particulate matter than modern filter-equipped diesel cars, Scientific Reports, 7, 4926, doi: 10.1038/s41598-017-03714-9, 2017.

P9 R24-30 CAMx performance in reproducing $SO_2$ are rather poor. Modelled concentrations are strongly overestimated (MFB is higher than 75% in February) and substantially uncorrelated to observed values (IOA is around 0.1). Considering that $SO_2$ emissions are mostly related to "Other anthropogenic sources" do authors think that such overestimation could influence also other species (e.g. NMVOC, PM) and, therefore, also the contribution of this sector to OM concentrations? (see for example conclusions P16 R5-9)

As we mentioned in P10 L10, the overestimation of $SO_2$ is most likely because we distributed all the emissions into the first layer. Unlike $NO_x$, PM and NMVOC, which are dominated by the area sources (road traffic for $NO_x$, biomass burning for PM, and solvent use for NMVOC), the major sources of $SO_2$ are point sources such as power plants (SNAP1) and industrial combustion (SNAP3 and 4) with stack emissions (Bieser et al., 2011). That is why a significant overestimation was only found for $SO_2$. For the organic aerosols, as most of the precursors and POA are from area emissions, the influence of the vertical distribution of emissions is small.

Bieser, J., Aulinger, A., Matthias, V., Quante, M., and Denier van der Gon, H. A. C.: Vertical emission profiles for Europe based on plume rise calculations, Environ. Pollut., 159, 2935-2946, doi: https://doi.org/10.1016/j.envpol.2011.04.030, 2011.

P10 R1-2 Table 2 is a bit misleading because it summarizes the model performance at all ACSM/AMS stations, which cover very different temporal periods. The different rows should be grouped somehow, for example separating stations covering the whole year from sites covering only winter periods,

summer periods, etc. Moreover, it could be useful adding, for each site, the number of available observations, as well as the observation period (though already reported in table I)

We agree with the referee that showing seasonal differences of the statistical results are important, but grouping the stations is very complicated as some sites covering more than one season while some only cover a short period in a specific season. To avoid any confusion, we added a column in Table 2 showing the observation period and number of available observations. The seasonal-based statistical results were also added in Table S5 following the general comment (2).

Technical corrections P19 R18 Table 1?

Corrected (should be on P11 R18?). One more sentence was added in P11 L31-L32 to make it more clear.

*"The modelled OA components were evaluated using AMS/ACSM measurements analyzed with PMF at different stations (Table 1). The statistical results are presented in Table 2."*

---

## Author Comment (AC2) · 28 Oct 2019

**Responses to the comments of referee #2**

We thank Dr. Manish Shrivastava for the valuable comments that have greatly helped us to improve the manuscript. Please find below our responses (in black) after the referee comments (in blue). The changes in the revised manuscript are written in *italic*.

Jiang et al. provide a modeling study of organic aerosols in Europe using a volatility basis set approach. This study is important for this region and provides good insights about sources and formation of OA. Results are evaluated with measurements including PMF analysis of AMS/ACSM data. Below I have several suggestions for improvement and also citation of relevant papers that need to be considered by the authors.

General comments on modified VBS approach: VBS is a framework that represents gas-particle partitioning and multigenerational aging of SOA. But depending on SIVOC emissions, reaction rates, functionalization/fragmentation branching etc. different implementations of VBS can produce very different results. Thus, it is important to describe VBS developments in the context of previous studies, specifically acknowledging and documenting differences. The authors describe their VBS as a modified VBS approach. But use of a "modified VBS" terminology has been used in 2 previous papers from M. Shrivastava et al. 2013, 2015. Those papers included both functionalization and fragmentation of organics and compared model results to several field measurements (surface based and aircraft measurements). See:
https://agupubs.onlinelibrary.wiley.com/doi/full/10.1002/jgrd.50160
https://agupubs.onlinelibrary.wiley.com/doi/full/10.1002/2014JD022563

The authors use the 1.5D VBS from Koo et al. To avoid confusion between author's version of VBS and previous 2 papers (above), I recommend the authors add a few sentences about how their modified VBS differs from the modified VBS aging parameterizations developed by M. Shrivastava et al. It may be better to refer to their VBS as 1.5D VBS, since this is what they used. It would be also instructive to compare their modified VBS results with those from M. Shrivastava et al. Note that Cholakian et al. 2018 (cited in this paper) used a similar modified VBS as Shrivastava et al. 2013, 2015.

Actually the term "modified" in this study has a different sense. Shrivastava et al. (2013; 2015) modified the VBS framework to include gas-phase fragmentation reactions and a nonvolatile (semisolid) SOA paradigm, while the objective of this study is to perform OA source apportionment in Europe with a VBS-based air quality model. Thus, the major modification we made is to extend the volatility sets with split sources, together with minor adjustment of parameters for the sets of new diesel vehicles and biomass burning based on chamber experimental data.

In order to distinguish the modified 1.5-D VBS from previous studies, we added the description about VBS development, as well as a comparison of our work with literature in section 2.2.

[revised manuscript text omitted]

Specific comments:

Page 3 Line 5-10: In addition to Hallquist et al. 2009, also cite M. Shrivastava et al. 2017 Review paper on SOA published in Reviews of Geophysics: https://agupubs.onlinelibrary.wiley.com/doi/full/10.1002/2016RG000540

Done

Page 3 Line 15-20: For WRF-Chem please cite 2 of the more recent papers on VBS implementation of SOA in addition to Shrivastava et al. 2011: https://www.nature.com/articles/s41467-019-08909-4

https://agupubs.onlinelibrary.wiley.com/doi/full/10.1002/jgrd.50160
Done

Page 7 Line 5: While several models underpredict OA from biomass burning, some models predict OA from biomass burning could be much more important. See https://agupubs.onlinelibrary.wiley.com/doi/full/10.1002/2014JD022563. This should be acknowledged here in addition to Hodzic et al. 2010 for biomass burning.

More references were added in P7 L16 regarding the underestimation of OA from biomass burning.

Shrivastava, M., Easter, R. C., Liu, X. H., Zelenyuk, A., Singh, B., Zhang, K., Ma, P. L., Chand, D., Ghan, S., Jimenez, J. L., Zhang, Q., Fast, J., Rasch, P. J., and Tiitta, P.: Global transformation and fate of SOA: Implications of low-volatility SOA and gas-phase fragmentation reactions, J. Geophys. Res.-Atmos, 120, 4169-4195, doi: 10.1002/2014jd022563, 2015.

Ciarelli, G., Aksoyoglu, S., El Haddad, I., Bruns, E. A., Crippa, M., Poulain, L., Äijälä, M., Carbone, S., Freney, E., O'Dowd, C., Baltensperger, U., and Prévôt, A. S. H.: Modelling winter organic aerosol at the European scale with CAMx: evaluation and source apportionment with a VBS parameterization based on novel wood burning smog chamber experiments, Atmos. Chem. Phys., 17, 7653-7669, doi: 10.5194/acp-17-7653-2017, 2017.

Jathar, S. H., Gordon, T. D., Hennigan, C. J., Pye, H. O. T., Pouliot, G., Adams, P. J., Donahue, N. M., and Robinson, A. L.: Unspeciated organic emissions from combustion sources and their influence on the secondary organic aerosol budget in the United States, Proceedings of the National Academy of Sciences, 111, 10473-10478, doi: 10.1073/pnas.1323740111, 2014.

Page 11: Aqueous chemistry of organic aerosols (OA) in fog can also increase OA by 4-20% (see Gilardoni et al. 2014 PNAS for Po Valley Italy measurements of aqueous SOA). Since the authors underestimate winter-time OOA, missing aqueous phase SOA in fog would be an important source. Although they are overestimating OA during the autumn due to modeled bias in relative humidity and wet scavenging, the high bias could be due to other reasons like overestimation of SIVOC emissions, biases in aging paramaterizations. This needs to be acknowledged as a caveat. The authors could also compare rain rates simulated by their model to measurements in that region to provide further evidence for model underestimation of rain rate/wet scavenging.

We thank the referee for comments on the possible explanations for model performance. The lack of aqueous-phase chemistry for SOA formation is definitely a potential reason for SOA underestimation. We added the explanation in P13 L19 – L21.

*"It could come from the missing pathways of SOA formation such as the aqueous processing of water-soluble organics (Ervens et al., 2011), which was found to contribute up to ~20% of winter OA measured in Bologna (Gilardoni et al., 2016; Meroni et al., 2017)."*

The overestimation of SIVOC emissions is also a possible explanation for overestimated OA in SPC. As we mentioned in P8 L26, "... *this factor (SVOC/POA ratio) presents substantial inter-country variability due to different combustion type, fuel parameters and operation conditions, indicating a potential over- or underestimation for a specific area by using the factor of 3 in the whole domain*". Since the calculation of SIVOC emissions is still with very high uncertainty, one can hardly conclude if it is over- or underestimated. That is why we attribute the major reason to the missing fog scavenging process in the model (which we can clearly observe from the comparison of measured and modelled relative humidity in Figure 2). However, we agree that it is important to highlight the potential effects of the highly uncertain SVOC emissions as a caveat for future studies. A general statement for all sites is added in Section 3.2, P13 L24-L27.

*"Another potential limitation is related to the uncertainties in SVOC/IVOC emissions. We adopted a factor of 3 for the SVOC and POA ratio for the whole domain, however, the substantial spatial and temporal variability of the factor could lead to over- or underestimation of SVOC emissions at site scale (Denier van der Gon et al., 2015) and therefore over- or underestimation of the SOA, as well as of POA."*

Both measured and modelled precipitation in Bologna and SPC during the observation time was very low (see the zoomed subplot below, total observed precipitation <3.8 mm for SPC and < 1 mm for Bologna). Although the model performance to reproduce such low precipitation amounts are not as

good as at other sites, we think the effects from such low precipitation amounts should be negligible. Scavenging at SPC was controlled by fog, as highlighted by the significant decrease of PM1 mass during night-time, in concomitance with the fog inset (Gilardoni et al., 2014). The lack of scavenging mechanisms in the model during night-time is evident in Figure 2, showing that the modeled OA is not able to follow the diurnal trend of the measured OA during the fog period.

[Figure]

Denier van der Gon, H. A. C., Bergström, R., Fountoukis, C., Johansson, C., Pandis, S. N., Simpson, D., and Visschedijk, A. J. H.: Particulate emissions from residential wood combustion in Europe – revised estimates and an evaluation, Atmos. Chem. Phys., 15, 6503-6519, doi: 10.5194/acp-15-6503-2015, 2015.

Ervens, B., Turpin, B. J., and Weber, R. J.: Secondary organic aerosol formation in cloud droplets and aqueous particles (aqSOA): a review of laboratory, field and model studies, Atmos. Chem. Phys., 11, 11069-11102, doi: 10.5194/acp-11-11069-2011, 2011.

Gilardoni, S., Massoli, P., Paglione, M., Giulianelli, L., Carbone, C., Rinaldi, M., Decesari, S., Sandrini, S., Costabile, F., Gobbi, G. P., Pietrogrande, M. C., Visentin, M., Scotto, F., Fuzzi, S., and Facchini, M. C.: Direct observation of aqueous secondary organic aerosol from biomass-burning emissions, Proceedings of the National Academy of Sciences of the United States of America, 113, 10013-10018, doi: 10.1073/pnas.1602212113, 2016.

Page 12 Above line 5: Instead of "biomass density" the authors could probably just say increasing biogenic emissions here?

Here we used *"biomass density"* to refer to the increased biomass leading to increased biogenic emissions. We rephrased the sentence in P12 L16 – L17 to avoid misunderstanding.

*"The OOA from biogenic sources (OOA-BIO) begins to increase from April, when the biogenic emissions increased with increasing temperature and biomass density"*

Page 12 Line 20: From Figure 6 it seems the authors could have applied a site specific scaling of POA emissions based on PMF HOA+BBOA. This could improve their POA, its diurnal variation and also IVOC emissions for biomass burning that are calculated as 4 times BB-POA. Please comment on use of a site-specific scaling of BB-POA and IVOC emissions based on PMF HOA+BBOA.

We totally agree that a site-specific scaling of POA would improve the model performance. However, the challenge for now is whether the coverage of OA measurements and PMF studies can support the site/region-specific POA emissions scaling and IVOC emissions for the whole domain. Meanwhile, we should note that the PMF studies are also associated with relevant uncertainties (as mentioned in P12 L20–L21). Developing a site-specific S/IVOC emission estimation method requires further improvements on measurement and analysis techniques, as well as more field measurements. We added a discussion about future works in section 3.2, P13 L24 – L31.

*"Another potential limitation is related to the uncertainties in SVOC/IVOC emissions. We adopted a factor of 3 for the SVOC and POA ratio for the whole domain, however, the substantial spatial and temporal variability of the factor could lead to over- or underestimation of SVOC emissions at site scale (Denier van der Gon et al., 2015) and therefore over- or underestimation of the SOA, as well as of POA. It could also partially explain the differences in model performance for the temporal variation of HOA and BBOA for each site. To further improve the model performance, it is necessary to continuously update the chemical mechanism in models by introducing the missing processes and improving the parameterization based on the advanced knowledge; as well as to improve the emissions by including more site-specific sources, IVOC and SVOC estimates, and updated diurnal variation profiles."*